# Equilibrium Reasoners: Learning Attractors Enables Scalable Reasoning

Benhao Huang [1]   Zhengyang Geng [1]   J Zico Kolter [1]

https://github.com/locuslab/EqR

## Abstract

Scaling test-time compute by iteratively updating a latent state has emerged as a powerful paradigm for reasoning. Yet, the internal mechanisms that enable these iterative models to generalize beyond memorized patterns remain fundamentally unclear. We hypothesize that such generalizable reasoning arises from learning *task-conditioned attractors*: a latent dynamical system where stable fixed points correspond to valid solutions. We formalize this process by introducing *Equilibrium Reasoners (EqR)*. EqR enables test-time scaling without relying on external verifiers or task-specific priors. Instead, our models scale internal dynamics along two axes: *depth* by running more iterations and *breadth* by aggregating stochastic trajectories from multiple initializations. Empirically, performance gains from scaling test-time compute are tightly coupled with better convergence to attractors. This attractor perspective allows neural networks to adaptively allocate test-time compute based on task difficulty. While simple cases converge within 1 to 5 iteration steps, the hardest cases benefit from massive test-time scaling. By unrolling up to an equivalent of *40,000* layers, this scalable latent reasoning boosts accuracy from 2.6% for feedforward models to over 99% on Sudoku-Extreme. We hope our attractor perspective sheds light on scalable reasoning.

## 1. Introduction

Scaling is a defining pattern of modern AI systems: accuracy improves with training data, model capacity, and, increasingly, *test-time compute*. Across settings ranging

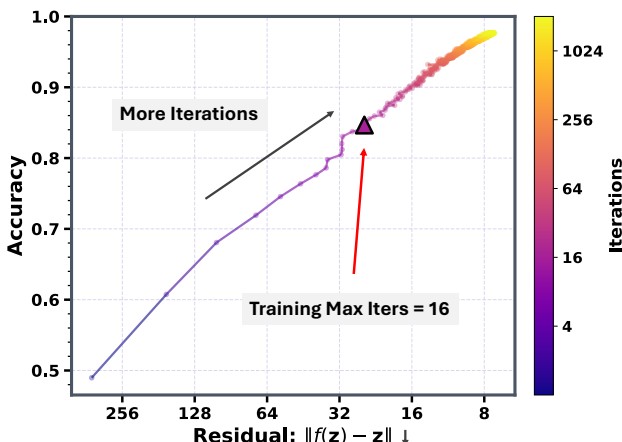

*Figure 1.* **Scalable test-time compute emerges from convergence to neural attractors.** We evaluate Equilibrium Reasoners (EqRs) with varying inference budgets and plot exact accuracy against the fixed-point residual $\|f_\theta(\mathbf{z}; \mathbf{x}) - \mathbf{z}\|$ (where $\mathbf{z}$ denotes latent states; lower indicates better convergence). Point color encodes the number of iterations on a log scale. Remarkably, while training is capped at 16 iterations, the learned attractor dynamics generalize to over 1,024 iterations at test time. Equivalent to unrolling over *40,000* effective layers, this massive scale-up drives the fixed-point residual down and boosts reasoning accuracy to over 99% on Sudoku-Extreme when further combined with breadth scaling via random initializations.

from search-based game agents (Silver et al., 2018) to chain-of-thought reasoning (Wei et al., 2022), systems often spend additional inference compute to improve performance.

Yet the opposite can be true, where more test-time compute may yield diminishing returns or even worse performance (Pipis et al., 2025; Ghosal et al., 2025; Fu et al., 2026; Chen et al., 2025). This suggests that improving reasoning via test-time scaling requires specific internal mechanisms. It raises a basic question: what internal mechanisms enable scalable and generalizable reasoning?

In this work, we study this problem on controlled, structured reasoning benchmarks, where memorization can be separated from generalization. Recent iterative reasoning models, such as HRM and TRM (Wang et al., 2025; Jolicoeur-Martineau, 2025), repeatedly apply a learned module to update latent states and achieve strong results on algorithmic

[1]Carnegie Mellon University, Pittsburgh, Pennsylvania, USA. Correspondence to: Zhengyang Geng <zhengyanggeng@gmail.com>.

*Proceedings of the $43^{rd}$ International Conference on Machine Learning*, Seoul, South Korea. PMLR 306, 2026. Copyright 2026 by the author(s).

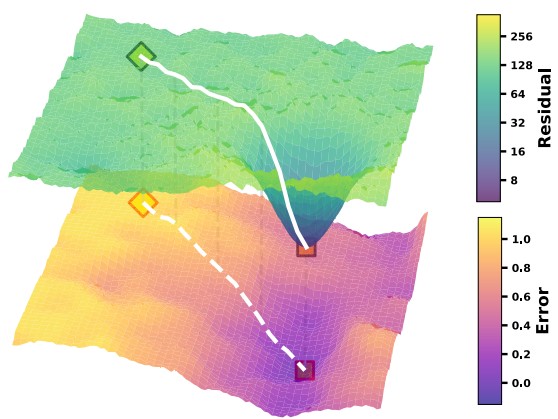

*Figure 2.* **Landscape alignment enables scalable and generalizable reasoning.** The lower surface depicts the task-metric landscape over solutions, while the upper surface depicts the model's learned internal landscape over latent states. We conceptualize EqR as learning a task-conditioned fixed-point system $\mathbf{z}^\star = f_\theta(\mathbf{z}^\star; \mathbf{x})$, whose attractors form an *internal landscape* in latent space. Training shapes this internal landscape so that its attractors align with the "attractor" of the *task-metric landscape* ("low-error basin"). By learning $f_\theta$, EqR models align basins across the two landscapes, so that descending the internal landscape via iterative updates also moves toward a correct task solution.

reasoning tasks such as Sudoku and Maze. This repeated update naturally defines a learned dynamical system in latent space for reasoning.

We argue that test-time scaling is effective when a model's *internal* attractor landscape aligns with the *task-metric* landscape: trajectories that achieve stronger convergence should also decode to lower-error answers. Under this view, training then seeks to shape the attractor landscape into a differentiable *surrogate* aligned with the task metric. **This amortizes intrinsically complex reasoning into a finite-capacity network, while leaving adaptive computation to inference.** Consequently, inference acts as an adaptive search: scaling up test-time compute reliably drives the latent state toward favorable attractors. In this aligned regime, finding stable fixed points (attractors) is implicitly solving the task, suggesting that stronger convergence yields better performance. The learned dynamics effectively close the capacity-complexity gap, enabling scalable reasoning that generalizes beyond memorization, even with limited data, model capacity, and training budgets.

We view these models as learned fixed-point dynamical systems (akin to a DEQ-style perspective (Bai et al., 2019)) whose trajectories evolve in latent space toward attractors. This extends the usual fixed-point view from asking whether a state converges to asking what attractor landscape the learned dynamics induce: which attractors exist, whether they are reachable from plausible initializations, and whether they align with the task metric. Under this

*Table 1.* **Weight-tied iteration is critical for generalization.** Exact accuracy (%) of standard feedforward models and weight-tied iterative models on Sudoku and Maze. Results marked with [†] are taken from prior work.

| Method | Sudoku (%) | Maze (%) |
|---|---|---|
| *Feedforward models* | | |
| 4-Layer Model | 1.8 | 0.0 |
| 16-Layer Model | 2.1 | 0.0 |
| 64-Layer Model | 2.6 | 0.0 |
| 256-Layer Model | *OOM* | *OOM* |
| *Iterative models (weight-tied)* | | |
| HRM (Wang et al., 2025) | 55.0[†] | 0.3 |
| TRM (Jolicoeur-Martineau, 2025) | 84.8 | 44.9 |
| URM (Gao et al., 2025) | 77.6[†] | 51.4 |
| **EqR (Ours)** | **99.8** | **93.0** |

lens, correct solutions correspond to *favorable* attractors and failures correspond to *spurious* attractors; scaling works when additional iterations or restarts guide trajectories into basins of favorable attractors.

Building on this view, we first perform a systematic study of training-time and inference-time design choices for iterative reasoning on controlled tasks, starting from the transition from feedforward computation to weight-tied iteration and then analyzing the training and inference choices that shape the learned attractor landscape. Across Sudoku-Extreme and Maze-Unique, we quantify convergence via the fixed-point residual $\|f_\theta(\mathbf{z}; \mathbf{x}) - \mathbf{z}\|$ and show that lower residual tightly tracks lower prediction error (Fig. 1), identifying the key factors that activate generalizable reasoning.

This framing suggests concrete, task-agnostic diagnostics: fixed-point convergence (a.k.a. $\|f_\theta(\mathbf{z}; \mathbf{x}) - \mathbf{z}\|$) and how it co-varies with prediction error. It also predicts a characteristic depth–breadth interaction: breadth (more restarts) becomes effective only after sufficient depth enables trajectories to meaningfully explore and settle into attractors, a pattern we observe in Fig. 3.

Guided by these diagnostics, we introduce two lightweight training interventions, randomized state initialization and path stochasticity via noise injection, to make favorable attractors easier to reach. The resulting dynamics can be scaled along two explicit inference axes: *depth* ($D$), the per-trajectory number of unrolled steps, and *breadth* ($B$), the number of stochastic trajectories from independent initializations. With two-axis scaled inference and convergence-based selection, EqR substantially outperforms prior iterative reasoning models on these controlled benchmarks, reaching 99.8% exact accuracy on Sudoku and 93.0% on Maze[1]. We hope these analyses help advance a mechanistic

---

[1] For simplicity, we refer to Sudoku-Extreme as Sudoku and Maze-Unique as Maze throughout the paper.

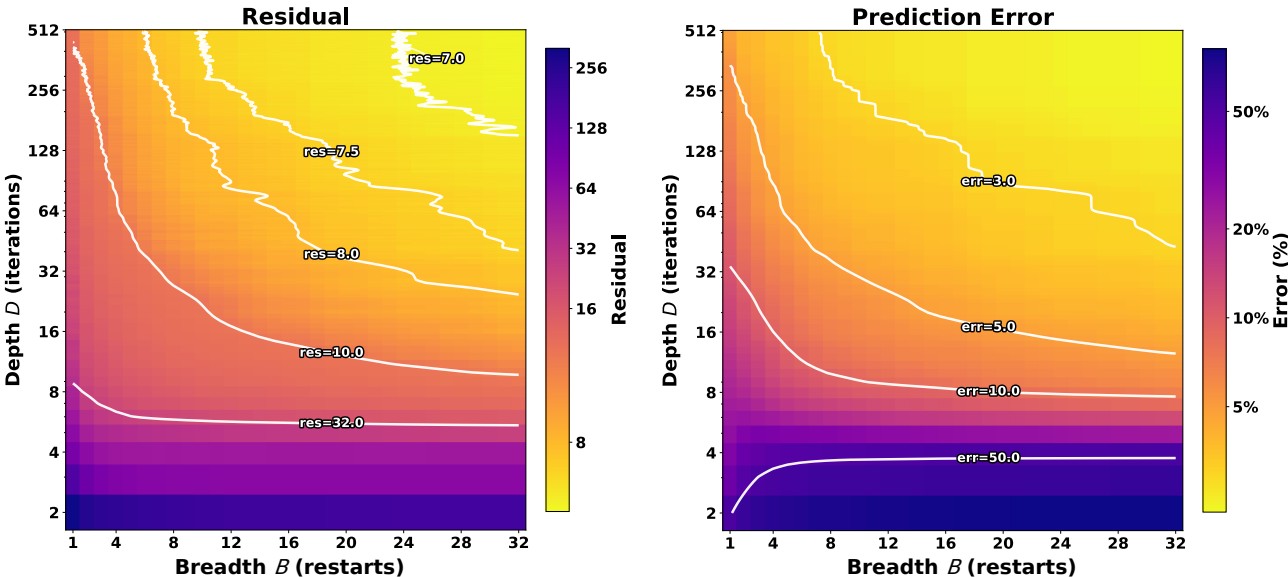

*Figure 3.* **Pareto heatmaps illustrating two-axis test-time scaling with iterations ("depth", $D$), and stochastic trajectories from multiple initializations ("breadth", $B$).** Left: fixed-point residual $r = \|f_\theta(\mathbf{z}; \mathbf{x}) - \mathbf{z}\|$ (power-law normalized), where $\mathbf{x}$ stands for model input and $\mathbf{z}$ for latent states. Right: prediction error (lower is better, log-scaled). Breadth scaling becomes effective only beyond a minimum number of model steps ($D \gtrsim 4$, or equivalently 168 layers unrolled), where more model steps allow restarts to better explore the landscape and find attractors. At sufficiently large "depth", increasing "breadth" consistently reduces fixed-point error and prediction error. Across the grid, lower prediction error is strongly associated with smaller residuals, supporting convergence as a reliable scaling signal.

understanding of iterative latent reasoning models and how their internal dynamics support scalable reasoning.

## 2. Background and Problem Formulation

We study *iterative reasoning models* that carry out multi-step computation through iterative updates of a latent state. Given an input $\mathbf{x} \in \mathcal{X}$, the model maintains a state $\mathbf{z}_k \in \mathbb{R}^n$ and applies a parameterized update operator

$$\mathbf{z}_{k+1} = f_\theta(\mathbf{z}_k; \mathbf{x}), \qquad (1)$$

where $k$ denotes the iteration index, and $\theta$ denotes the model parameters. This update-rule perspective is common in neural networks with iterative computation (Bai et al., 2019; Dehghani et al., 2019; Zhu et al., 2025; Wang et al., 2025; Jolicoeur-Martineau, 2025; Hao et al., 2025). These approaches share a common view of test-time scaling through additional applications of the same learned update rule. Starting from an initial state $\mathbf{z}_0 \sim \mu_0(\cdot \mid \mathbf{x})$, the model runs $K$ updates and decodes the final state into a prediction $\hat{\mathbf{y}}_K$. The task supplies a metric comparing $\hat{\mathbf{y}}_K$ with the target $\mathbf{y}$. In this work, we focus on iterative reasoning models with fixed-size latent states. Hierarchical Reasoning Models (Wang et al., 2025) and Tiny Recursive Models (Jolicoeur-Martineau, 2025) implement multi-step reasoning by iteratively updating high- and low-level latent states in a nested-loop schedule. For our analysis, the essential commonality is a weight-tied latent dynamical system whose extra computation unfolds in state space.

## 3. From Feedforward Predictors to Iterative Reasoners

We study how a feedforward model can be turned into an iterative model (Wang et al., 2025; Jolicoeur-Martineau, 2025) with feedback loops under controlled data and compute, and use this path to analyze the key design choices for training strong iterative models. We ablate the main design axes used by prior works as follows.

**Weight-tied structure.** The weight-tied design reuses parameters across model layers, replacing additional distinct layers with repeated iterations of the same update block.

**Truncated gradients.** Full backpropagation through long weight-tied trajectories is costly and multiplies many recurrent Jacobians, which can make the backward dynamics poorly conditioned and the gradient signal unstable. Truncated gradients with detached carry keep the length of forward trajectory but cut the backward graph at segment boundaries. Therefore, each update is optimized through a local trajectory window.

**Hierarchical iterations.** We then compare single-stream iteration against hierarchical iterations, where two latent states are updated at different frequencies. This separates the effect of weight-tied iteration from the additional two-timescale structure used in HRM/TRM-style models.

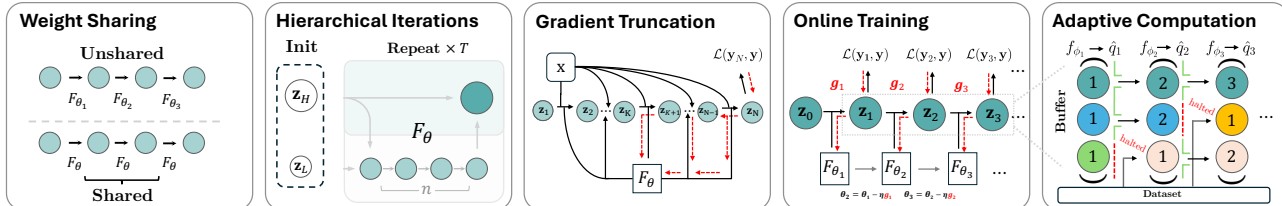

*Figure 4.* **Controlled construction path.** We build a controlled path from a feedforward model to a scalable iterative reasoner by progressively adding five ingredients: (1) **Weight-tied parameters** across iteration steps; (2) **Gradient truncation** with detached carry to stabilize optimization through long trajectories and reduce cost; (3) **Segmented online training** to shape intermediate solver states through interleaved parameter updates along the latent-state trajectory; (4) **Hierarchical iterations** with coupled fast/slow latent updates; and (5) **Adaptive computation** to allocate compute by problem difficulty. Together, these components turn a feedforward model into a stable iterative reasoning process whose trajectories can converge to solution-bearing attractors.

**Supervision placement and optimization schedule.** After choosing the gradient window, one must decide where losses are placed and when parameters are updated. Given a $K$-step trajectory $\{\mathbf{z}_k\}_{k=1}^{K}$ from iterative models, we compare three schedules: *1) Vanilla*, which computes the loss only after the final iteration and updates the parameters once per full trajectory; *2) Trajectory Supervision*, which places losses at multiple iterations but accumulates them into a single update at the end of the trajectory; and *3) Segmented Online Training*, which splits the trajectory into segments, supervises the end of each segment, and takes an optimizer step immediately. The next segment starts from the current latent state with detached carry, but under the updated parameters. Thus, SOT changes not only where supervision is applied, but also the optimizer time scale: the model is updated along the evolving trajectory rather than only after the full rollout has completed. From an optimization viewpoint, this can be seen as an alternating approximation to an attractor-learning problem: latent updates seek a reachable low-residual state under the current operator, while parameter updates reshape the operator so that these reachable states decode to correct solutions. These schedules can differ substantially in optimization fidelity, training stability, and efficiency. We include detailed discussions in Appendix A.2.

**Adaptive computation time (ACT).** We also study adaptive computation via a learned halting mechanism (Graves, 2017). Let $\hat{q}_k = f_\phi(\mathbf{z}_k)$ be a halting score and $\tau = \min\{k \leq K : \hat{q}_k > \delta\}$, with $\tau = K$ if no halt is triggered. We compare different variants, including fixed-depth iteration, oracle halting, and learned halting with an ACT head. The main distinction is whether the halting signal is only predicted or is actually used to allocate variable compute. In the latter case, solved examples leave the batch early while unresolved examples receive further refinement, so ACT acts as a difficulty-aware compute allocation mechanism.

**Overview.** Together, these axes define the construction path studied in Sec. 6.1: 1) weight-tied structure converts distinct layers into repeated application of a shared update

block; 2) hierarchical iterations test whether two-timescale latent updates add benefits beyond single-stream iteration; 3) truncated gradients stabilize optimization through long weight-tied trajectories by keeping the backward graph local while also reducing memory and compute costs; 4) segmented online training changes where supervision and optimizer updates enter the trajectory; and 5) adaptive computation reallocates iteration budget across examples by difficulty. The main text reports the compact construction path, and we defer full details, results and diagnostics to Appendix A.2.

## 4. Iterative Models as Attractor Dynamics

This section develops the conceptual framework used by the rest of the paper. We first relax exact fixed-point convergence into an attractor view of iterative inference, then use the resulting landscape modes to explain when depth and breadth scaling should help and what training must shape.

### 4.1. From Fixed-Point Convergence to Attractors

Prior iterative reasoning work already points toward a convergence interpretation: HRM describes its nested latent updates through *hierarchical convergence*, while TRM cautions that literal fixed-point convergence is too strong because latent residuals can remain nonzero even as they decrease during training (Wang et al., 2025; Jolicoeur-Martineau, 2025). By contrast, we argue that iterative models do converge in a weaker attractor sense: repeated application of the update operator often reduces the residual $\|f_\theta(\mathbf{z}; \mathbf{x}) - \mathbf{z}\|$ and improves performance, as illustrated in Fig. 1. The key point is that this behavior need not be exact fixed-point convergence. Under finite computation, a trajectory may approach a fixed point, settle into a stable region, or enter a bounded recurrent set; when nearby states are drawn toward such a set under repeated updates, it acts as an attractor. We therefore use *attractor* to describe stable long-run outcomes of the learned dynamics, generalizing the equilibrium perspective used in Deep Equilibrium Models (Bai et al., 2019).

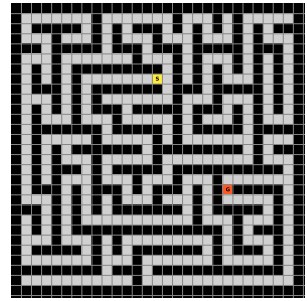
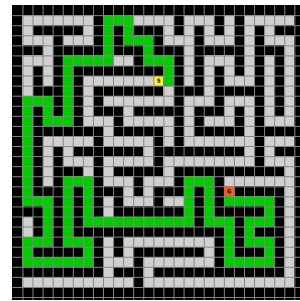

*Figure 5.* **Datasets and Evaluations.** We evaluate our models and baselines on *Sudoku-Extreme*, a challenging $9 \times 9$ Sudoku benchmark for long-horizon constraint satisfaction, and on *Maze-Unique*, a uniquely solvable variant of Maze-hard-1k designed to remove ambiguity caused by multiple shortest paths. Additional details are provided in Appendix C.

This attractor view keeps the core convergence claim without requiring convergence to a single exact fixed point: test-time compute is useful when trajectories move toward favorable attractors and lower-residual states within a well-structured *internal landscape*. A favorable basin suffices.

In this view, the learned trajectory is part of the prediction mechanism: when successive states become more task-consistent as their residuals fall, additional iterations can refine the answer instead of simply adding compute. Feedforward models do not induce such refinement trajectories; in our controlled comparison, they generalize substantially worse than iterative alternatives in Tab. 1.

Formally, for a data example $(\mathbf{x}, \mathbf{y}) \sim \mathcal{D}$, inference induces a trajectory $\{\mathbf{z}_k\}_{k \geq 0}$ by iterating an update operator $f_\theta(\cdot; \mathbf{x})$ from the initialization $\mathbf{z}_0 \sim \mu_0(\cdot \mid \mathbf{x})$. We write $\mathcal{Z}_\theta^*(\mathbf{x})$ for the stable long-run outcomes of these dynamics (e.g., fixed points or small recurrent sets).

The collection $\mathcal{Z}_\theta^*(\mathbf{x})$ is the model's *attractor landscape*. This landscape matters through two axes: task alignment and reachability. Task alignment asks whether the reached attractors decode to correct solutions rather than spurious ones. Reachability asks which attractor a trajectory reaches, and how reliably it does so under different initializations or perturbations. We summarize reachability using *breadth* and *depth*: broad attractors are easy to reach from many initial states, while deep attractors are stable once reached.

These two geometric properties map directly to two test-time scaling levers. *Depth scaling* increases the number of forward iterations $D$, giving a single trajectory more opportunities to refine within the basin it has entered. *Breadth scaling* runs $B$ independent restarts from initial states $\{\mathbf{z}_0^{(i)}\}_{i=1}^B$ and aggregates their outputs, increasing coverage over possible basins. We use the number of function evaluations, $\text{NFE} = D \cdot B$, as a compact way to describe inference budgets throughout the scaling experiments.

### 4.2. Landscape Modes and Scaling Implications

The attractor landscape view becomes useful when it predicts how test-time compute should be allocated. Fig. 7 in the appendix visualizes how task alignment, reachability, and the two scaling levers combine into four qualitative regimes. Each regime identifies the dominant failure source and therefore predicts whether depth scaling, breadth scaling, or neither should help. [2]

(a) **No correct attractor:** all reachable attractors decode to poor task outcomes. The failure is task misalignment rather than insufficient compute, so residual reduction does not translate into task improvement and neither depth nor breadth scaling helps.

(b) **Correct and spurious attractors coexist:** a correct attractor exists, but inference may converge to competing low-residual, high-error attractors. The failure is basin selection, so breadth scaling is most useful because additional restarts increase the chance of entering the correct basin; depth helps only after the trajectory enters that basin.

(c) **Correct but hard to reach:** the correct attractor is nearly unique but has a narrow or weak basin. The failure is reachability, so breadth increases the chance of entering the basin and depth can help weakly attracted trajectories settle once they do; gains are limited by basin mass and stability.

(d) **Well-aligned landscape:** the correct attractor is broad and stable, so residual decay is tightly coupled with task-error reduction. Depth reliably refines trajectories toward the solution, while breadth provides additional coverage but is no longer the main bottleneck.

---

[2]Task error is defined at the sequence level: any token mismatch counts as incorrect. Token-level losses therefore induce a qualitatively different and much noisier landscape than the task-level metric in this setting, so we visualize only the latter.

Thus, depth and breadth are complementary: depth refines a trajectory after it reaches a useful basin, while breadth increases basin coverage, consistent with the depth–breadth interaction in Fig. 3. Test-time scaling succeeds when correct attractors are both aligned and reachable, motivating the training interventions in Sec. 5.

# 5. Shaping Attractor Landscapes

Sec. 4.2 shows that test-time scaling is effective when the learned landscape contains correct attractors and inference reaches them reliably. **Attractor landscape shaping** is therefore the guiding training principle: we want the iterative dynamics to (i) admit correct solutions as stable attractors and (ii) make their basins easy to reach from diverse initial states as test-time compute increases. We now describe how to move generic iterative models toward **Equilibrium Reasoners**.

We introduce two task-agnostic interventions that do not require external verifiers or hand-crafted search heuristics: (1) randomized state initialization (RI), which samples initial latent states rather than model weights to improve coverage under breadth scaling and reduce train–test mismatch, and (2) noise injection (NI), which implements path stochasticity by perturbing each iteration step to mitigate premature trapping and broaden exploration as the iteration budget grows. Pseudocode is provided in Algorithm 1 in the appendix.

## 5.1. Randomized State Initialization

HRM and TRM (Wang et al., 2025; Jolicoeur-Martineau, 2025) typically train with a fixed initial state $\mathbf{z}_0$ shared across trajectories. In contrast, we sample $\mathbf{z}_0 \sim \mu_0(\cdot \mid \mathbf{x})$ independently for each trajectory. This matches training to breadth scaling at test time, where multiple draws of $\mathbf{z}_0$ probe different basins. The benefit is twofold: it broadens the regions shaped during training and encourages stable predictions across restarts.

**(i) Coverage of correct attractors.** With a fixed initializer, learning is constrained to a small state-space neighborhood and tends to shape trajectories only locally. This limits exposure to alternative basins. Randomizing $\mathbf{z}_0$ expands the explored region during training and increases the likelihood that correct attractors are reachable at inference.

**(ii) Stability and path independence.** Randomizing $\mathbf{z}_0$ also promotes consistency across restarts: the same $(\mathbf{x}, \mathbf{y})$ is observed under multiple initial states, so divergent predictions are penalized. This encourages path independence (Anil et al., 2022) by aligning predictions across trajectories.

By default, we use a Gaussian $\mu_0(\cdot \mid \mathbf{x})$ with covariance $\sigma_0 I$. Appendix A.3 and Appendix A.3 study learnable initializers and initialization scale; we use a fixed Gaussian initializer in the main experiments to isolate stochastic coverage from learned-prior design.

## 5.2. Path Stochasticity via Noise Injection

Random initializations reduce the train–test gap induced by breadth scaling; path noise regularizes *how* trajectories evolve. This targets modes **(b)** and **(c)** in Sec. 4.2: mild noise can help trajectories enter better basins and avoid premature convergence to incorrect stable states. Thus, RI and NI act on complementary parts of the trajectory: RI broadens where rollouts start, while NI smooths the local dynamics encountered along each rollout.

We augment the iteration with damping and additive noise:

$$\mathbf{z}_{k+1} = \mathbf{z}_k + (1 - \lambda)\, r_\theta(\mathbf{z}_k; \mathbf{x}) + \beta\, \varepsilon_k, \qquad (2)$$

where $\varepsilon_k \sim \mathcal{N}(0, I)$. Here $\lambda \in [0, 1)$ controls damping and $\beta \geq 0$ controls the noise magnitude. In other words, we inject isotropic Gaussian noise at each step and use $\beta$ to control its strength. This preserves the same update architecture while allowing controlled local exploration around the deterministic trajectory. We consider variants with different $\beta$ and a learnable noise variant in Appendix A.4. Empirically, mild damping ($\lambda = 0.05$) combined with small path noise ($\beta = 0.01$) performs best among the tested variants. At test time, one can increase $\beta$ under breadth scaling to foster exploration, analogous to temperature scaling.

# 6. Experiments

We organize the experiments around two questions. First, we ask what ingredients turn a feedforward model into a strong iterative model. Second, based on the iterative backbone, we test whether landscape-shaping interventions improve accuracy and make depth and breadth scaling reliable.

**Task representation.** Each puzzle is serialized into a token sequence. The input sequence encodes the unsolved puzzle, and the target sequence encodes its solution, as illustrated in Fig. 5. The sequence length is fixed during inference for a given task, since each task uses a fixed grid size, but it differs across tasks, e.g., Sudoku ($9 \times 9$) versus Maze ($30 \times 30$). See more details in Appendix C.

**Evaluation metrics.** By default, i.e., without breadth scaling, we report *exact accuracy*, which equals 1 only if all tokens are correct and 0 otherwise. Under breadth scaling with $B$ independent restarts, we consider three evaluation metrics in this paper: *(1) Averaged exact accuracy*, the mean exact-match accuracy over the $B$ restarts ($B{=}1$ reduces to the standard single run); *(2) Top-1 convergence accuracy*, which selects the restart with the smallest average residual over the final few iterations ($L{=}3$) and checks whether its prediction is correct; and *(3) Majority-vote accuracy*, which

*Table 2.* **A clean construction path on Sudoku-Extreme.** Starting from a feedforward MLP-mixer predictor, each row adds the next ingredient that incrementally leads to strong iterative models. The **Blocks** column denotes the number of distinct trainable blocks; **NLE** denotes the per-trajectory number of equivalent layer evaluations at the reported evaluation point; breadth scaling is not used here ($B=1$).

| Methods | Blocks | Param.(M) | NLE | Train Acc. | Eval Acc. |
|---|---|---|---|---|---|
| vanilla feedforward | 42 | 105.6 | 42 | 93.8 | 2.6 |
| + weight-tied | 2 | 5.03 | 42 | 94.5 | 32.6 |
| + SOT + depth ×16 | 2 | 5.03 | 672 | 94.9 | 74.7 |
| + hierarchical recurrence | 2 | 5.03 | 672 | 99.3 | 76.5 |
| + ACT training | 2 | 5.03 | 672 | 82.2 | **84.8** |

predicts by majority vote across restarts. Formal definitions are given in Appendix D.3.

**Baselines.** For Sec. 6.1, we begin with a feedforward baseline and then introduce iterative components incrementally. For Sec. 6.2, we also report HRM (Wang et al., 2025) and TRM (Jolicoeur-Martineau, 2025) baselines without our training interventions or inference extrapolation. For the TRM comparison, the backbone is matched at the block level: we use TRM's task-specific settings for Sudoku and Maze, and each block has a task-dependent token mixer followed by an MLP. The mixer is an MLP-mixer on Sudoku and self-attention on Maze. This makes the comparison about the training and inference changes on a comparable iterative backbone, instead of a new block architecture. The feedforward baselines use 42 blocks on Sudoku and 15 blocks on Maze; the corresponding weight-tied variants use 2 blocks with 21 iterations and 1 block with 15 iterations, respectively. See Appendix D.1 for full details.

### 6.1. From Feedforward Models to Iterative Models

We study the gain from turning a feedforward model into an iterative one in this section. Tab. 2 starts from a vanilla feedforward predictor and adds weight-tying, long unrolls, hierarchy, and ACT.

Tab. 2 shows a monotonic construction path. Compared with the feedforward baseline under the same budgets, weight-tied models show significant improvement (from 2.6% to 32.6%); when we further scale up the depth with the supervision and update schedule needed to train long unrolls, the performance improves to 74.7%[3]. Hierarchical recurrence provides a smaller additional gain in this setting, ACT training allows difficulty-aware allocation of training-time compute per sample, and further improves inference performance to 84.8%. The ACT row has lower training accuracy because the model learns to stop early to prevent overfitting on easier examples while allocating more computation to

---

[3]Scaling the depth of weight-tied models requires a well-designed supervision and optimization schedule. We include the details in Appendix A.2.

*Table 3.* **Learning favorable attractor landscapes.** Effect of the proposed training interventions: randomized state initialization (**RI**) and noise injection (**NI**; path stochasticity). Evaluated at the base compute budget ($D=16, B=1$).

| Method (Landscape Shaping) | Sudoku | Maze |
|---|---|---|
| baseline | 84.8 | 44.9 |
| + train w/ RI | 86.0 | 68.6 |
| **+ train w/ RI + NI (EqR)** | **86.4** | **82.2** |

*Table 4.* **Test-time compute scaling.** Exact accuracy (%) under depth scaling (outer iterations $D$) and breadth scaling (restarts $B$) for EqR. Depth scaling increases the per-trajectory rollout length, whereas breadth scaling increases the number of independent trajectories. We use $D$ and $B$ for these two axes and defer the full compute accounting to Appendix D.4.

| Test-Time Strategy | $D$ | $B$ | Sudoku | Maze |
|---|---|---|---|---|
| EqR baseline | 16 | 1 | 86.4 | 82.2 |
| + depth scaling | 64 | 1 | 93.0 | 88.9 |
| + breadth scaling | 64 | 128 | **99.8** | **93.0** |

learn harder ones, which gives the best evaluation accuracy in the construction path. Full ablations, update-schedule variants, ACT diagnostics, FLOPs accounting, and supporting figures are in Appendix A.2.

**Summary and takeaway.** Overall, the evidence supports a specific training principle: Weight-tied models create iterative capacity, but realizing that capacity at larger depth requires well-designed training strategies to shape the dynamics so that late-iteration states stay aligned with the task objective under finite stability and memory constraints.

### 6.2. From Iterative Models to Equilibrium Reasoners

Training with the proposed interventions (Sec. 5) improves both accuracy and the reliability of test-time scaling, consistent with shaping a more favorable attractor landscape. Across tasks, we observe three consistent effects: (i) higher baseline accuracy without additional test-time compute, (ii) a higher scaling ceiling as inference uses more depth, with further gains when depth is combined with breadth, and (iii) stronger empirical alignment between residual convergence and task correctness, which makes convergence-based selection effective under breadth scaling.

In Tab. 3, baseline denotes the final TRM-style iterative model from Tab. 2, before applying randomized state initialization or noise injection. We compare our methods against baselines on Sudoku and Maze in Tab. 3 and Tab. 4.

**❙ Landscape shaping improves accuracy and stability without extra inference compute.**

At the training-time compute budget, models trained with randomized state initialization and noise injection consistently outperform the same backbone trained without these

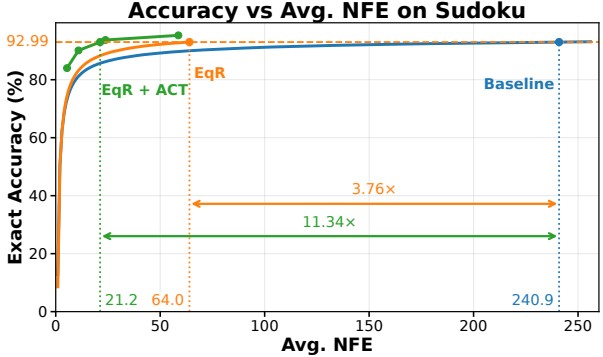

*Figure 6.* **Test-time depth-scaling efficiency on Sudoku-Lite.** At a matched exact-accuracy target of 92.99%, EqR requires 3.76× fewer NFEs than the baseline, and EqR+ACT improves this to 11.34× fewer NFEs. The gain is therefore not explained by increased test-time compute alone: the proposed training interventions make the same accuracy reachable with substantially less inference compute.

interventions. In Tab. 3, the full RI+NI training improves Sudoku from 84.8 to 86.4 and Maze from 44.9 to 82.2, with RI alone already raising Maze accuracy to 68.6. These gains hold on both training and evaluation splits, suggesting that correct attractors become reachable for a larger fraction of inputs within the same compute budget (Sec. 4.2). The same interventions also improve inference stability, as reflected by stronger path independence (Tab. 11 in the appendix, Appendix A.4).

▎**Shaped landscapes raise the test-time scaling ceiling.**
After applying the landscape-shaping interventions, increasing depth first improves within-trajectory refinement, and combining the deeper rollout with breadth scaling further improves coverage across restarts. Tab. 4 makes this effect explicit: at $B=1$, increasing the depth from $D=16$ to $D=64$ raises EqR from 86.4 to 93.0 on Sudoku and from 82.2 to 88.9 on Maze. Combining this deeper rollout with breadth scaling ($B=128$) further increases accuracy to 99.8 on Sudoku and 93.0 on Maze. This pattern suggests that landscape shaping supports both forms of test-time scaling: deeper rollouts improve a single trajectory, while broader restarts cover more attractor basins. This is consistent with the landscape modes shown in Fig. 7 in the appendix: interventions primarily mitigate modes **(b)** and **(c)** by enlarging correct basins and reducing spurious trapping.

▎**Convergence becomes a reliable selection signal after landscape shaping.**
Majority voting and convergence-based selection are both breadth-scaling strategies, but they exploit different signals. Majority voting aggregates *decoded* outputs, whereas convergence-based selection picks the run with the strongest convergence signal, e.g., the smallest average residual over the final few iterations. After learning a favorable landscape, convergence aligns better with task correctness, so selecting

the *Top-1 Converged* run becomes a reliable and compute-efficient selection rule. We visualize this trend on Sudoku in Fig. 8 in the appendix: as breadth increases, Top-1 Converged achieves comparable or higher expected accuracy than majority vote for the same number of restarts, with our training interventions.

This selection rule is not universally valid: for the baseline TRM, residual reduction can indicate convergence to spurious attractors, so Top-1 Converged can underperform majority voting. It becomes reliable only after shaping the landscape so residual convergence tracks task error. This distinction is important: convergence is useful here not as a task-agnostic certificate, but as a learned proxy whose reliability depends on the attractor landscape.

**Summary.** Overall, the results suggest that the proposed interventions improve baseline accuracy and stability, raise the test-time scaling ceiling across depth and breadth, and strengthen convergence–correctness alignment. As a result, convergence becomes a more reliable selection signal under breadth scaling, offering a more efficient alternative to majority voting once the landscape is well shaped. Additional experiments in Appendix A.5 show that the same recipe also improves Mini-ARC performance and transfers to a Transformer backbone.

### 6.3. Adaptive Computation for Budget-Elastic Inference

So far, our scaling experiments have applied the same test-time budget to every task instance, regardless of difficulty. However, applying a massive, static compute budget universally is highly inefficient. As our landscape analysis suggests, instance difficulty is highly heterogeneous: simple problems quickly fall into favorable attractors, whereas complex ones require extensive iterative refinement. To systematically optimize the compute-accuracy Pareto frontier, we resort to elastic budget inference guided by a learned halting policy (Jolicoeur-Martineau, 2025). By equipping EqR with a learned halting head, the model dynamically tracks its internal state and terminates computation early upon converging to an attractor, so additional compute is allocated mainly to instances that remain unresolved.

We use a fixed-size inference queue for batching: halted samples are replaced immediately to keep utilization high while preserving per-sample stopping. This allows ACT to convert sample-wise dynamic halting into an actual reduction in the average number of function evaluations (Avg. NFE) at inference time.

Tab. 5 shows that ACT substantially reduces Avg. NFE with only minor accuracy changes: at $D=1024$, Avg. NFE drops from 1024.0 to 58.7 (17.4× fewer evaluations) while accuracy changes from 96.1 to 95.3; under breadth scaling (64, 128), Avg. NFE drops from 8192.0 to 1400.6 (5.8×

*Table 5.* **EqR with Adaptive Computation Time (ACT) on Sudoku-Lite.** We report exact accuracy (%) and Avg. NFE.

| Budget $(D, B)$ | ACT | Eval Acc.↑ | Avg. NFE↓ |
|---|---|---|---|
| (16, 1) | ✗ | 84.3 | 16.0 |
| | ✓ | 84.0 | **5.4** |
| (64, 1) | ✗ | 90.9 | 64.0 |
| | ✓ | 90.1 | **10.9** |
| (256, 1) | ✗ | 94.6 | 256.0 |
| | ✓ | 93.7 | **23.8** |
| (1024, 1) | ✗ | 96.1 | 1024.0 |
| | ✓ | 95.3 | **58.7** |
| (64, 128) | ✗ | 97.9 | 8192.0 |
| | ✓ | 97.4 | **1400.6** |

fewer evaluations) while accuracy changes from 97.9 to 97.4. This suggests most instances terminate early, while a small fraction require long runs. Fig. 6 shows the same effect at the accuracy–compute frontier: EqR+ACT reaches the matched accuracy target with $11.34\times$ fewer NFEs than the baseline.

Together with earlier scaling results, this spans both ends of compute: large-budget scaling and budget-elastic efficiency. The halting objective's training-side effect was analyzed in Tab. 8(e) in the appendix; here we focus on ACT's inference-side compute–accuracy trade-off.

## 7. Related Work

**Iterative weight-tied models.** Iterative models apply an update operator repeatedly to refine a latent state, with representative works including weight-tied Transformers such as the Universal Transformer and related variants (Dehghani et al., 2019; Graves, 2017; Chowdhury & Caragea, 2025; Heo et al., 2025). Deep Equilibrium Models (Bai et al., 2019) take this idea to the implicit limit by defining representations as fixed points, followed by extensive work on convergence diagnostics, stability, and more efficient training methods (Bai et al., 2021; Geng et al., 2021a;b; Gu et al., 2020; Anil et al., 2022; Fung et al., 2022) and practical tooling such as TorchDEQ (Geng & Kolter, 2023).

Path-independent equilibrium models make this connection more explicit: when iterative inference converges to the same fixed point regardless of the trajectory or initialization, additional test-time steps can reliably refine toward a well-defined representation (Anil et al., 2022). Our setting relaxes the requirement of a globally unique fixed point: we instead ask whether finite rollouts and restarts concentrate around solution-aligned attractors, making path independence both a diagnostic and a property encouraged by our intervention.

Recent weight-tied models further show that iterative latent computation is becoming an active scaling direction across language and visual reasoning (Geiping et al., 2025; Zhu et al., 2025; Prairie et al., 2026; Song et al., 2026; Bae et al.,

2025; Shu et al., 2026). In parallel, the HRM series of works shows that such iterative models have strong performance over complex and structured reasoning tasks (Wang et al., 2025; Jolicoeur-Martineau, 2025; Gao et al., 2025). These finite iterative computations still induce a latent dynamical system; we study whether training can shape their trajectories toward solution-aligned attractors, enabling scalable gains from additional test-time compute.

**Compute allocation in weight-tied models.** For weight-tied models, test-time compute allocation asks how many iterations to spend, where to spend them, and whether the same budget should be used uniformly or adapted across inputs. One line of work studies iteration or recurrent-depth scaling, showing that applying a weight-tied module more times at test time can improve performance beyond the training regime (Schwarzschild et al., 2021; Geiping et al., 2025; Prairie et al., 2026; Shu et al., 2026). A second line focuses on adaptive allocation, using mechanisms such as token-wise depth (Song et al., 2026), learned recurrence mixtures (Bae et al., 2025), adaptive halting (Graves, 2017), or elastic-depth budget conditioning (Jeddi et al., 2026) to spend different amounts of computation across tokens, examples, or compute budgets. Recent compute-allocation diagnostics (Moosa et al., 2026) and selective latent iterations on hard tokens (Fu et al., 2025) further ask whether adaptive policies allocate compute to genuinely hard tokens rather than merely reducing average depth. These model-specific mechanisms connect to a broader inference-time scaling lesson: additional computation is most useful when allocated to informative candidates, states, or search branches rather than spent uniformly (Snell et al., 2025; Yang et al., 2025; Ghosal et al., 2025). In our work, we use depth scaling and breadth scaling to verify that scaling far beyond the training regime can substantially improve performance. We then close the efficiency side by learning a halting module following Graves (2017), which cuts inference cost by allocating fewer iterations to easier inputs while preserving most of the scaling gains.

## 8. Conclusion

We present an attractor-based view of test-time scaling in iterative reasoning models. Trajectory diagnostics show that depth and breadth help when learned attractors align with the task metric and are reachable from diverse initial states. Randomized initialization and path noise reshape this landscape, improving correct-attractor coverage and stability and making extra computation reliable. This suggests evaluating iterative latent reasoning models not only by final accuracy, but also by whether their dynamics make correct solutions stable, reachable, and selectable. Together, these diagnostics and interventions move toward a mechanistic account of scalable iterative reasoning.

## Impact Statement

This work studies mechanisms for making iterative latent reasoning models more reliable under test-time scaling. Its main positive impact is methodological: attractor diagnostics can help distinguish useful computation from convergence to incorrect states. The experiments use controlled reasoning benchmarks and do not deploy a high-stakes system. If similar methods are used in broader decision-making pipelines, failures may be harder to detect, so we emphasize benchmark specificity and careful evaluation across task difficulty.

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

# Appendix

## A. Additional Results, Analyses, and Findings

This section extends the findings in the main text through additional analyses and experiments, including a residual-diagnostic formulation, training-dynamics ablations from feedforward models to weight-tied iterative models, ablations of training interventions on initialization and path stochasticity, generalization beyond the main setting, and seed stability.

### A.1. Attractor Formulation and Residual Diagnostics

This subsection isolates the attractor formulation and residual diagnostics used to interpret the training ablations below.

A useful idealization of attractor learning follows the DEQ view of an input-conditioned equilibrium layer (Bai et al., 2019):

$$\min_{\theta,\mathbf{z}} \ \ell_\theta(\mathbf{z};\mathbf{x},\mathbf{y}) \qquad \text{s.t.} \qquad R_\theta(\mathbf{z};\mathbf{x}) = \mathbf{z} - f_\theta(\mathbf{z};\mathbf{x}) = 0. \tag{3}$$

The constraint says that the supervised latent state should be a fixed point of the current update operator. This differs from the Universal Transformer formulation (Dehghani et al., 2019), where input token embeddings initialize the recurrent state and the shared block is iterated over that state. Here the update is written as an input-conditioned solver $f_\theta(\mathbf{z};\mathbf{x})$: the problem data $\mathbf{x}$ remains available as an external condition at every solver step, as in DEQs and recent recurrent-depth models (Geiping et al., 2025). In our setting, however, the relevant object is not a globally unique fixed point, but a reachable attractor or low-residual state reached from an initialization $\mathbf{z}_0$ by iterative computation. We can therefore write the practical surrogate as a bilevel optimization problem:

$$\min_{\theta} \quad \mathbb{E}_{(\mathbf{x},\mathbf{y}),\mathbf{z}_0}\big[\ell_\theta(\mathbf{z}_\theta^\star(\mathbf{x},\mathbf{z}_0);\mathbf{x},\mathbf{y})\big]$$
$$\text{s.t.} \quad \mathbf{z}_\theta^\star(\mathbf{x},\mathbf{z}_0) := \text{Solve}_\theta^{(D)}(\mathbf{z}_0;\mathbf{x}), \quad \|\mathbf{z}_\theta^\star - f_\theta(\mathbf{z}_\theta^\star;\mathbf{x})\|_2 \le \varepsilon_{\text{res}}. \tag{4}$$

where $\ell_\theta(\mathbf{z};\mathbf{x},\mathbf{y})$ denotes the supervised loss evaluated from latent state $\mathbf{z}$, $\text{Solve}_\theta^{(D)}$ denotes a $D$-step lower-level rollout induced by repeated applications of $f_\theta$, and $\varepsilon_{\text{res}}$ is the residual tolerance for the reached state.

**Fixed-point residual as a local convergence diagnostic.** The formulation clarifies why residual is a meaningful diagnostic under local stability. Suppose the reached basin contains a fixed point $\mathbf{z}^\star = f_\theta(\mathbf{z}^\star;\mathbf{x})$, and $f_\theta$ is $L$-Lipschitz in that basin for some $L < 1$. Then

$$\|\mathbf{z} - \mathbf{z}^\star\| \le \|\mathbf{z} - f_\theta(\mathbf{z};\mathbf{x})\| + \|f_\theta(\mathbf{z};\mathbf{x}) - f_\theta(\mathbf{z}^\star;\mathbf{x})\|$$
$$\le \|R_\theta(\mathbf{z};\mathbf{x})\| + L\|\mathbf{z} - \mathbf{z}^\star\|, \tag{5}$$

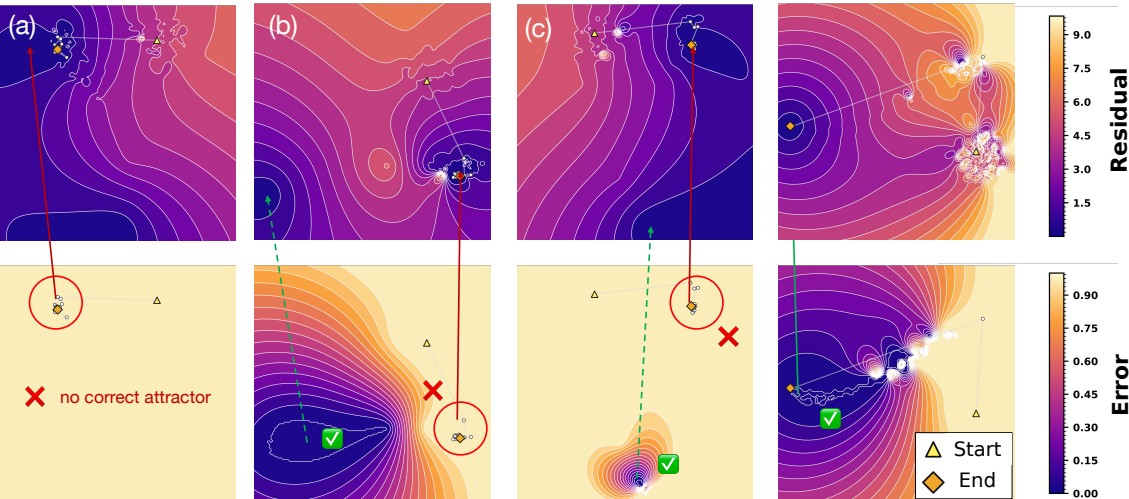

*Figure 7.* **Four attractor landscape modes visualized by residual and task error.** We run 512 random initializations over 256 Sudoku-Extreme examples, project trajectories to 2D with PCA, and color by sequence-level error: **(a)** no correct attractor; **(b)** correct and spurious attractors; **(c)** a narrow correct basin; **(d)** a broad, mostly unique correct attractor.

so $\|\mathbf{z} - \mathbf{z}^\star\| \le \|R_\theta(\mathbf{z}; \mathbf{x})\|/(1 - L)$. Thus, inside a stable basin, small residual implies closeness to the local attractor. To connect residual to correctness, one also needs an output margin. Let $s_{\theta,i,a}(\mathbf{z}; \mathbf{x})$ be the logit for assigning label $a$ at output location $i$, and suppose $\mathbf{z}^\star$ decodes to the correct output $\mathbf{y}$. Define the minimum margin at the attractor by

$$\gamma(\mathbf{z}^\star) = \min_i \left[ s_{\theta,i,y_i}(\mathbf{z}^\star; \mathbf{x}) - \max_{a \ne y_i} s_{\theta,i,a}(\mathbf{z}^\star; \mathbf{x}) \right]. \tag{6}$$

If $\gamma(\mathbf{z}^\star) > 0$ and each true-versus-competing logit gap is $G_{\text{gap}}$-Lipschitz in the same basin, then states within distance $\gamma(\mathbf{z}^\star)/G_{\text{gap}}$ decode correctly. Combining this margin condition with Eq. 5, a sufficient residual condition is

$$\|R_\theta(\mathbf{z}; \mathbf{x})\| < (1 - L)\frac{\gamma(\mathbf{z}^\star)}{G_{\text{gap}}} \quad \implies \quad \hat{\mathbf{y}}_\theta(\mathbf{z}; \mathbf{x}) = \mathbf{y}. \tag{7}$$

Thus residual is a correctness proxy only under local stability, a correct attractor, and positive output margin; low residual near a spurious or low-margin attractor still certifies convergence, not correctness.

**Implicit gradient conditioning.** The same formulation also clarifies why exact implicit gradient computation through long attractor solvers can be unstable. Let $J_{\mathbf{z}} = \partial_{\mathbf{z}} f_\theta(\mathbf{z}^\star; \mathbf{x})$ and $J_\theta = \partial_\theta f_\theta(\mathbf{z}^\star; \mathbf{x})$. At an exact locally isolated fixed point, differentiating $R_\theta(\mathbf{z}^\star; \mathbf{x}) = 0$ for fixed $\mathbf{x}$ gives

$$(I - J_{\mathbf{z}})\, d\mathbf{z}^\star = J_\theta\, d\theta, \qquad \frac{d\mathbf{z}^\star}{d\theta} = (I - J_{\mathbf{z}})^{-1} J_\theta, \tag{8}$$

with Jacobians evaluated at $(\mathbf{z}^\star, \theta, \mathbf{x})$. Thus, solver sensitivity is controlled by the resolvent $(I - J_{\mathbf{z}})^{-1}$: when this operator is poorly conditioned, small parameter changes can cause large attractor shifts, and approximation errors in the lower-level solve can be amplified in the implicit gradient. This conditioning issue is a standard concern in implicit-layer training, where prior work studies Jacobian regularization, monotone or well-posed operators, and approximate or Jacobian-free backward passes (Bai et al., 2021; Geng et al., 2021a;b; Gu et al., 2020; Fung et al., 2022). The role of truncation and SOT is therefore not only to reduce memory and compute, but also to keep optimization local while the latent trajectory tracks a changing attractor landscape.

## A.2. From Feedforward Models to Iterative Models: Training-Dynamics Ablations and Diagnostics

This subsection expands the construction path from feedforward models to weight-tied iterative models in Sec. 6.1, Tab. 2, using the design axes introduced in Sec. 3, on Sudoku tasks. The full ablations cover weight-tied structure, hierarchical iterations, gradient truncation, supervision and optimization schedules, adaptive computation variants, and FLOPs and memory accounting. For weight-tied models, we follow the configurations of TRM listed in Tab. 17. For a fair comparison,

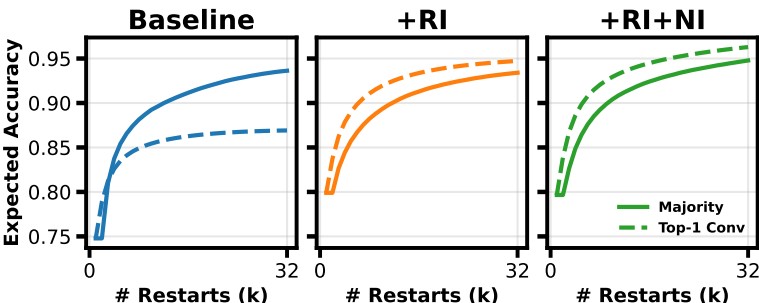

*Figure 8.* **Breadth-scaling efficiency of aggregation rules on Sudoku.** Expected accuracy versus the number of restarts for majority vote and Top-1 Converged across TRM, TRM+RI, and EqR (TRM+RI+NI). After applying RI, especially RI+NI, Top-1 Converged becomes more compute-efficient than majority vote and reaches a higher accuracy ceiling.

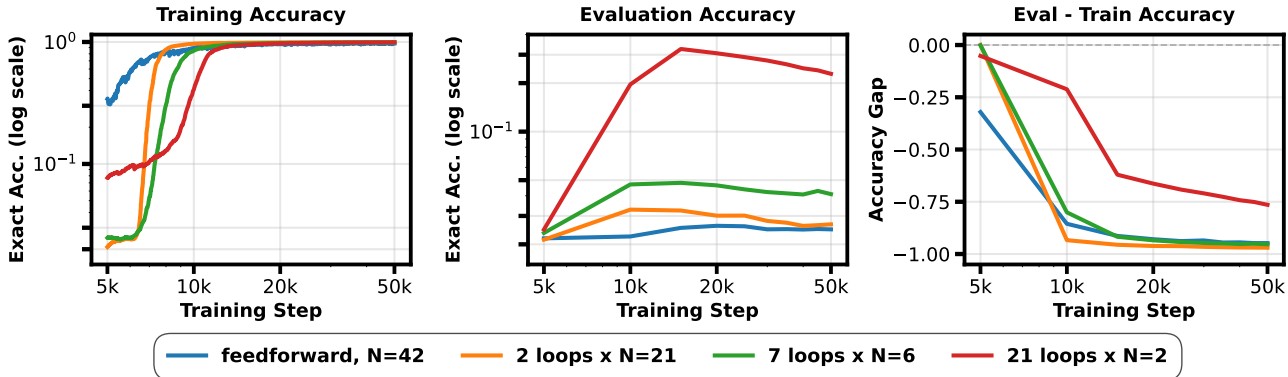

*Figure 9.* **Iterative depth improves generalization under a matched layer-evaluation budget.** On Sudoku, increasing the iteration steps changes both optimization and generalization: the shallow iterative model reaches higher training accuracy than the feedforward depth baseline at its best evaluation checkpoint, while deeper iterative models trade slightly lower training accuracy for substantially higher evaluation accuracy under the same approximate layer-evaluation budget. The three panels plot training accuracy, evaluation accuracy, and the evaluation-minus-training accuracy gap. The axis choices expose different scales of variation: the left and middle panels use log-scaled vertical axes for positive training and evaluation accuracy, while the right panel keeps a padded linear gap axis because evaluation-minus-training accuracy is signed; all panels use log-scaled training steps to emphasize early fitting dynamics. Curves start at 5k training steps, matching the first available evaluation checkpoint.

the Sudoku feedforward baseline uses 42 distinct blocks, matching equivalent-layer budget $N_{eq} = 42$ of the corresponding weight-tied iterative model (Appendix D.4). Throughout this subsection, an *iteration* means one outer-loop step or segment, which may itself contain repeated applications. We count these through equivalent-layer evaluations. A trajectory with $D$ iterations contains $D$ outer-loop segments, and an iteration-depth multiplier scales the number of segments rather than the individual layer applications inside each segment.

### Iterative models are fundamentally different from feedforward models.

Tab. 8(a) shows that introducing a weight-tied iterative model improves substantially over the vanilla feedforward baseline on Sudoku (2.6%→32.6%); Fig. 9 further shows the corresponding training/evaluation error curves on Sudoku, under matched layer-evaluation budgets. The curves show a more nuanced trade-off than a simple train–evaluation gap: at its best-evaluation checkpoint, the 2-iteration weight-tied model reduces both training and evaluation error relative to the feedforward depth baseline; among iterative models, increasing the iteration steps then raises the training error while reducing the evaluation error. Tab. 6 records this checkpoint-level comparison for the four plotted runs.

Thus, a weight-tied parameterization creates useful iterative capacity: under the matched layer-evaluation budget, the feedforward model underperforms in both training and evaluation, whereas replacing independent layers with repeated applications of shared parameters yields better generalization and a smaller train–evaluation mismatch. However, this capacity remains insufficient under the current budget, motivating the depth-scaling and additional training-dynamics ablations below.

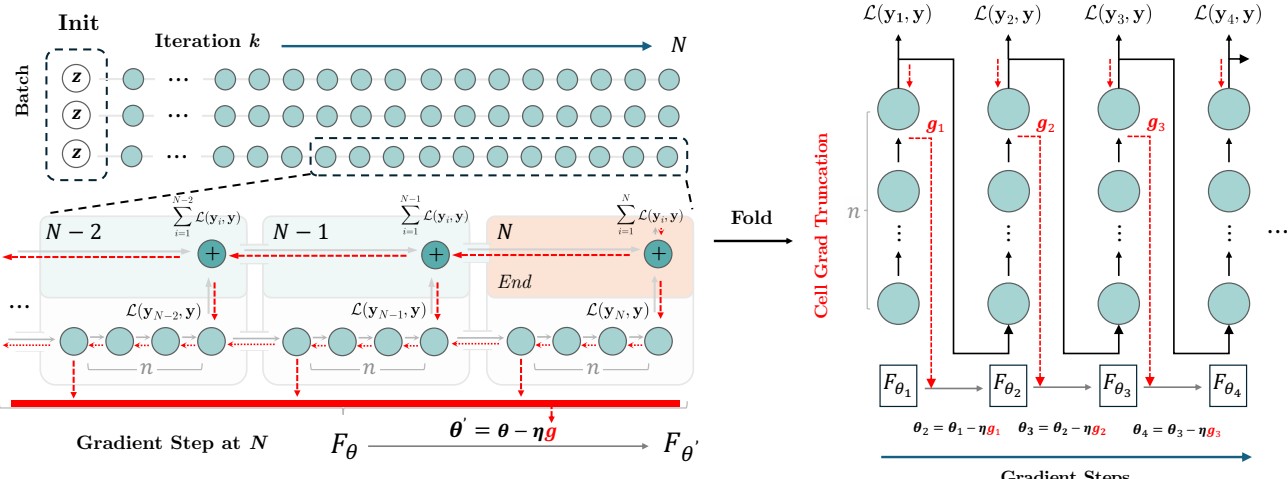

*Figure 10.* **Gradient flow in segmented online training.** Left: a segment loss supervises the current solver state while backpropagating only through the immediately preceding local computation graph, marked by the red dashed region. Right: over a long trajectory, SOT changes the optimizer time scale: each segment performs a local backward pass and an immediate parameter update, then the next segment continues from the detached carried state under the updated shared parameters. This lets training follow long iteration trajectories without backpropagating through the entire history, while still providing local credit assignment to the states that produced each supervised segment.

*Table 6.* **Training error at the checkpoint of best evaluation accuracy for Fig. 9.** Because training and evaluation are logged at slightly different steps, the training error is taken from the nearest available training checkpoint to the best-evaluation checkpoint.

| Model | Layer-eval budget | Best eval acc. (%) | Best eval error (%) $\downarrow$ | Train error near best eval (%) $\downarrow$ |
|---|---|---|---|---|
| feedforward | $1 \times 42$ | 2.6 | 97.4 | 6.2 |
| 2 iters. | $2 \times 21$ | 3.3 | 96.7 | 1.8 |
| 7 iters. | $7 \times 6$ | 4.8 | 95.2 | 3.6 |
| 21 iters. | $21 \times 2$ | 32.6 | 67.4 | 3.8 |

### ▍Iteration steps induce a trade-off between alignment and feasibility.

Larger iterations (depth) are desirable because they can move states closer to stable, solution-aligned attractors; however, they also make training less feasible. As shown in Tab. 8(b), doubling the iteration depth is already helpful, improving Sudoku from 32.6% to 51.3%. The natural next step is to scale this trajectory to a much larger depth multiplier (e.g., $16\times$), but the full-gradient backpropagation through the long trajectory is memory-prohibitive and recurrent-gradient products can explode or vanish. Detaching the carried state before the terminal loss makes the $16\times$ run feasible: the backward graph covers only the final local transition, and with a stronger learning rate and lower weight decay this terminal-loss recipe with detached carry reaches 51.8%. However, the gain over the $2\times$ setting is marginal relative to the additional computation, indicating that simply extending the terminal-loss trajectory with detached carry is not an efficient way to leverage the potential of long iterations. This motivates a more careful choice of supervision placement and optimization schedule.

The local residual and implicit gradient conditions behind this interpretation are separated in Appendix A.1; here we focus on the training-dynamics evidence.

### ▍Segmented online training alternates latent-state and parameter updates.

The weak gain from terminal-loss training with detached carry suggests that feasibility is not the only issue: once gradients are detached along most of the trajectory, the intermediate carried states receive little direct supervision. A natural compensation is therefore to add loss anchors along the long trajectory, so that more carried states are trained while each backward graph remains local. As introduced in Sec. 3, we call this trajectory supervision (an offline deep-supervision schedule): the model evaluates losses at multiple carried states,

$$\mathcal{L}_{\text{off}}(\theta) = \frac{1}{M} \sum_{m=1}^{M} \ell_\theta(\mathbf{z}_{t_m}; \mathbf{x}, \mathbf{y}), \qquad \mathbf{z}_{k+1} = f_\theta(\mathbf{z}_k; \mathbf{x}), \tag{9}$$

and updates the parameters only after the full trajectory has been processed. This keeps training cost controlled and helps

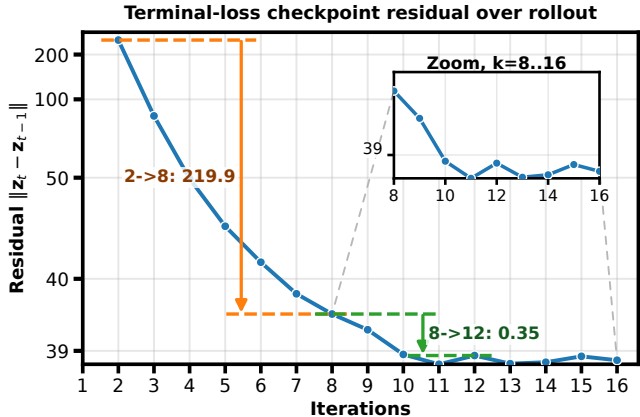

*Figure 11.* **Residual diagnostic for late anchors.** At the terminal-loss detached-carry checkpoint, the rollout residual $\|\mathbf{z}_t - \mathbf{z}_{t-1}\|$ drops sharply in the early iterations and then changes only slightly over the late trajectory. The shifted-log scale keeps this late plateau visible: the change from iterations 8 to 12 is small compared with the early 2 to 8 drop, supporting the use of late anchors that supervise states after the trajectory has largely stabilized. Together with Tab. 7, this separates early transient anchors, which can conflict with the rollout dynamics, from late anchors placed near a stable basin.

restore supervision along the trajectory.

However, offline trajectory supervision creates a stale-trajectory mismatch. With parameters held fixed while anchors are collected, several transient lower-level iterates are all asked to match the same upper-level target before the operator has been updated. The resulting objective can behave like average-state matching: it rewards moving an aggregate of those states toward the target, even when that aggregate lies off the trajectory reachable by the updated operator. Thus, extra anchors provide more local supervision, but they can also pull the learned dynamics toward a target that is not reachable under the updated operator.

SOT instead alternates the two levels. For a segment of $h$ function evaluations, $\mathbf{z}_s$ denotes the detached carried state from the previous segment, while the current segment remains gradient-tracked:

$$
\begin{aligned}
\tilde{\mathbf{z}}_{s+1}(\theta) &= f_\theta^{(h)}(\mathbf{z}_s; \mathbf{x}), \\
g_s &= \nabla_\theta \ell_\theta(\tilde{\mathbf{z}}_{s+1}(\theta); \mathbf{x}, \mathbf{y})\big|_{\theta=\theta_s}, \\
\theta_{s+1} &= \theta_s - \eta g_s, \qquad \mathbf{z}_{s+1} = \text{stopgrad}(\tilde{\mathbf{z}}_{s+1}(\theta_s)).
\end{aligned}
\tag{10}
$$

The next latent segment is then generated under the updated parameters $\theta_{s+1}$. Each segment first takes a lower-level corrector step in latent space, then takes an upper-level parameter step that reshapes the operator. This alternating view is natural because changing $\theta$ changes the attractor landscape, while advancing $\mathbf{z}$ reveals which basin the current operator can actually reach. If the $h$-step local solver contracts errors by a factor $\rho < 1$ near the current attractor and the local attractor map has sensitivity $\kappa_\theta \approx \|(I - J_{\mathbf{z},s})^{-1} J_{\theta,s}\|$, where $J_{\mathbf{z},s}$ and $J_{\theta,s}$ are the corresponding local Jacobians at segment $s$, then the carried-state tracking error $e_s = \|\mathbf{z}_s - \mathbf{z}_{\theta_s}^\star\|$ obeys the bound

$$
\begin{aligned}
e_{s+1} &= \|f_{\theta_s}^{(h)}(\mathbf{z}_s; \mathbf{x}) - \mathbf{z}_{\theta_{s+1}}^\star\| \\
&\leq \|f_{\theta_s}^{(h)}(\mathbf{z}_s; \mathbf{x}) - \mathbf{z}_{\theta_s}^\star\| + \|\mathbf{z}_{\theta_s}^\star - \mathbf{z}_{\theta_{s+1}}^\star\| \\
&\lesssim \rho e_s + \kappa_\theta \|\theta_{s+1} - \theta_s\|.
\end{aligned}
\tag{11}
$$

This separates the two effects: latent correction reduces the first term, while the parameter update contributes the attractor shift in the second term. Compared with trajectory supervision under detached carry, SOT therefore keeps supervision closer to the currently reachable trajectory instead of optimizing many stale anchors before any parameter update occurs.

The diagnostics below test this interpretation. They first show that early anchors can conflict with the detached trajectory, and then show that SOT turns the same long-rollout supervision into a much more effective training procedure.

Together, Fig. 11 and Tab. 7 show that intermediate anchors are not uniformly harmful: anchors over the full trajectory include early transient states and reduce accuracy from 51.8% to 47.1% relative to terminal-loss supervision, whereas anchors restricted to the late trajectory can improve performance. We interpret the late-anchor gain through the attractor-alignment view: if solution alignment is mediated by attractor basins, the same target loss has a different meaning before and after the trajectory reaches the basin. The residual trace gives a concrete proxy for this split: after the sharp early drop, later states lie on a more stable part of the rollout, so supervising them creates less conflict with the model's own dynamics.

*Table 7.* **Late-anchor supervision ablation.** Detached $16\times$ trajectory; accuracy in %.

| Supervision | Anchors | Acc. |
|---|---|---|
| terminal loss | 16 | 51.80 |
| full anchors | 1:16 | 47.10 |
| late anchors | 8:16 | 51.36 |
| late anchors | 12:16 | 57.50 |

Before the trajectory enters a solution-aligned basin, supervision on intermediate states mainly acts as coarse guidance, and its local gradients can be unreliable because they are attached to transient states rather than to a stable solution. After the trajectory enters the basin, the same target supervision becomes more trustworthy because the local state is already near the solution attractor. Late anchors therefore concentrate gradient signal in the regime where the gradients are aligned with the desired attractor. Under this view, if several late anchors all lie in the attractor basin, accumulating or upweighting those late losses behaves like a larger effective step size on reliable gradients, instead of amplifying noisy early-trajectory gradients. For each anchor range in Tab. 7, we report the best accuracy over learning rates $\{10^{-3}, 5\times10^{-4}, 10^{-4}, 5\times10^{-5}\}$ and weight decays $\{0.1, 0.5, 1.0\}$.

The SOT rows give the direct evidence for the alternating formulation. As shown in Tab. 8(c), switching from trajectory supervision at $16\times$ to SOT at $16\times$ improves Sudoku from 47.1% to 74.7% under the same nominal depth multiplier. The gain is therefore not explained by adding anchors alone: the key change is that parameter updates are interleaved with latent-state updates, so later trajectory segments are generated by the current operator rather than by stale parameters. SOT can further reduce memory and compute cost when combined with in-segment gradient truncation, which shortens the backward graph inside each segment. Tab. 8(d) shows that this truncation interacts strongly with the latent structure. In the single-latent setting, in-segment truncation reduces Sudoku-Extreme accuracy from 74.7% to 67.2%. With hierarchical iterations, however, the same truncation improves accuracy from 69.8% to 75.4%. This reversal suggests that hierarchical iterations shape the training dynamics differently from the single-latent update, an interaction we discuss in the next finding.

### ▌Hierarchical iterations change performance, but their effect is difficult to decouple from the training recipe.

Hierarchical iterations interact with other training strategies. As shown in Tab. 8(d), without in-segment truncation, the single-latent model is stronger on Sudoku, whereas with truncation the hierarchical variant becomes stronger. The Adaptive Computation Time (ACT) ablation rows in Tab. 8(e) show a second interaction with the halting mechanism. With learned ACT, the hierarchical latent model reaches 84.8% on Sudoku, while the corresponding single-latent $\mathbf{z}$ variant reaches 73.9%. Thus, hierarchy is not a standalone switch; its effect depends on the surrounding training recipe. We also observed that the relative performance of hierarchical iterations compared with the single-latent model depends on the task.

### ▌Adaptive Computation Time (ACT) changes training dynamics through learned halting.

The halting mechanism is not only for efficiency: when applied to the training of iterative models, it also shapes the training dynamics, and different halting signals lead to different outcomes. As shown in Tab. 8(e), oracle (ground-truth-based) halting collapses the hierarchical model from 75.4% to 13.6% on Sudoku, whereas a learned ACT head improves the result from 75.4% without it to 84.8% in this setting. Furthermore, we find that training a learned ACT head, even when the predicted halting signal is not used for dynamic early exit during training, can mitigate overfitting, as shown in Fig. 12 (■ Training Head + No Early Halt vs. ■ No Early Halt).

This hierarchy dependence is one reason we report ACT as part of the training-dynamics recipe rather than as a purely inference-side add-on.

### ▌Cost accounting explains why long-trajectory training is feasible.

Tab. 9 gives the symbolic cost accounting behind these choices. The key distinction is between the length of the forward trajectory and the length of the backward graph. Let $T$ denote the number of outer-loop steps covered by a long trajectory. We write $C(\cdot)$ for the backward and parameter-update cost per optimizer interval and $M(\cdot)$ for the retained training memory

| Model | Blocks | Iter. | Acc. |
|---|---|---|---|
| ① feedforward | 42 | 1 | 2.6 |
| ② weight-tied | 2 | 21 | 32.6 |

*(a)* Feedforward vs. weight-tied

| Recipe | Acc. |
|---|---|
| ② weight-tied | 32.6 |
| ③ + 2× depth | 51.3 |
| ④ + 16× depth | *OOM* |
| ⑤ + terminal loss | 51.8 |

*(b)* Depth scaling limits

| Recipe | Acc. |
|---|---|
| ⑤ terminal loss + detached carry | 51.8 |
| ⑥ trajectory supervision | 47.1 |
| ⑦ segmented online training | 74.7 |

*(c)* From final-only to SOT

| Latents | w/ grad | Acc. |
|---|---|---|
| ⑦ $\mathbf{z}$ | 21 | 74.7 |
| ⑧ $\mathbf{z}$ + trunc. | 7 | 67.2 |
| ⑨ $(\mathbf{z}_L, \mathbf{z}_H)$ | 21 | 69.8 |
| ⑩ $(\mathbf{z}_L, \mathbf{z}_H)$ + trunc. | 7 | 75.4 |

*(d)* Latents and truncation

| Halting | Latents | Acc. |
|---|---|---|
| ⑩ ✗ | $(\mathbf{z}_L, \mathbf{z}_H)$ | 75.4 |
| ⑪ GT | $(\mathbf{z}_L, \mathbf{z}_H)$ | 13.6 |
| ⑫ ✓ | $(\mathbf{z}_L, \mathbf{z}_H)$ | **84.8** |
| ✓ | $\mathbf{z}$ | 73.9 |

*(e)* ACT halting

*Table 8.* **Training dynamics ablations.** 2× and 16× denote iteration-depth multipliers. The subtables enumerate the recipe choices used in the training-dynamics analysis. Subtable (a) compares a 42-block feedforward solver with a 2-block, 21-iteration weight-tied solver under a matched equivalent-layer budget. Subtable (b) varies the terminal-loss trajectory length and whether the long carried state is detached before the terminal loss. Subtable (c) compares two loss/update schedules on the long detached trajectory: trajectory supervision accumulates segment losses before one optimizer update, while segmented online training (SOT) applies an optimizer update after each segment and then continues from the carried state. Subtable (d) factors the SOT rows by latent structure (single $\mathbf{z}$ versus hierarchical $(\mathbf{z}_L, \mathbf{z}_H)$) and by in-segment gradient truncation, which shortens the backward window. Subtable (e) varies the halting mechanism for comparison. We report the best performance after tuning learning rates and weight decay for each variant. All accuracies are reported as percentages. FLOPs are reported separately in Tab. 9.

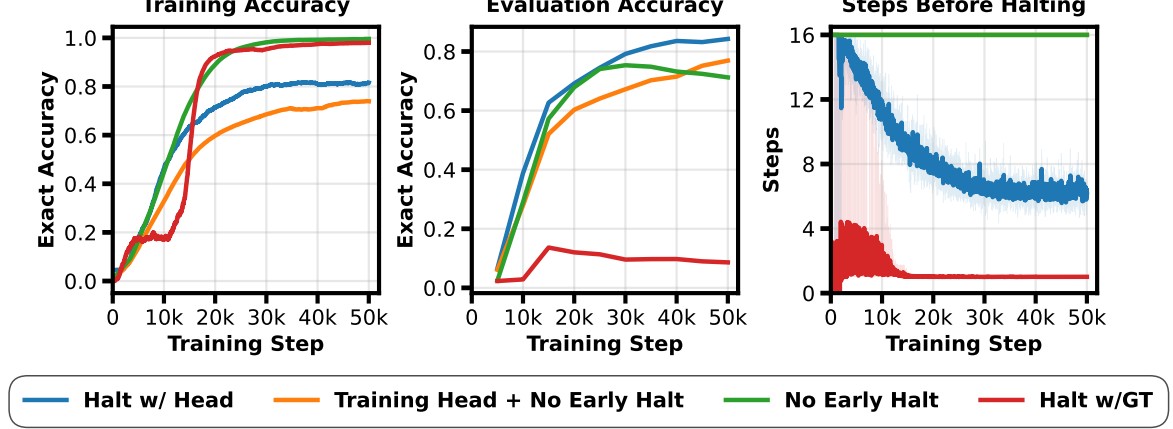

*Figure 12.* **Ablation of ACT halting mechanisms.** We compare several ACT variants (including oracle halting based on ground-truth correctness) and report both task performance and the resulting average number of iteration steps at 50k training steps; oracle halting tends to overfit and collapses toward near-single-step behavior.

per optimizer interval. The resulting accounting separates the effects of detached carry and truncation, which reduce backward and memory pressure, from SOT, which changes when the optimizer update happens. A more detailed derivation of this symbolic cost accounting is available in a separate blog post (Huang, 2026). For the remaining local terms, we use:

| *Backward and parameter-update costs* | |
|---|---|
| $c_\ell$ | One local loss/head backward. |
| $c_B$ | Segment parameter-backward. |
| $c_B^{\text{trunc}}$ | Truncated segment parameter-backward. |
| $c_J$ | Temporal state-backward through a segment. |
| $c_\theta$ | Extra shared-gradient accumulation. |
| $c_u$ | Parameter update for one recurrent block. |

| *Retained memory terms* | |
|---|---|
| $a_f$ | Activation memory for one segment. |
| $a_\ell$ | Activation memory for one loss/head branch. |
| $a_f^{\text{det}}$ | Segment activation after detached carry. |
| $a_f^{\text{trunc}}$ | Segment activation under truncation. |
| $P$ | Parameter-side memory for one recurrent block. |

These terms come from the local loop backward equations. For one segment transition $\mathbf{z}_{s+1} = f_\theta(\mathbf{z}_s; \mathbf{x})$, define

$$J_s = \frac{\partial \mathbf{z}_{s+1}}{\partial \mathbf{z}_s}, \qquad B_s = \frac{\partial \mathbf{z}_{s+1}}{\partial \theta}, \tag{12}$$

$$\bar{\mathbf{z}}_s = J_s^\top \bar{\mathbf{z}}_{s+1}, \quad \nabla_\theta^{(s)} \mathcal{L} = B_s^\top \bar{\mathbf{z}}_{s+1},$$

where the second line uses column adjoints and $\bar{\mathbf{z}}_{s+1} = d\mathcal{L}/d\mathbf{z}_{s+1}$ is the incoming adjoint. Thus $c_J$ denotes the temporal state-backward vector–Jacobian product through $J_s$, while $c_B$ denotes the segment parameter-backward vector–Jacobian product through $B_s$. For shared recurrent weights, the step-local parameter contributions $\nabla_\theta^{(s)} \mathcal{L}$ are accumulated into a shared gradient buffer, which gives the $c_\theta$ term. These equations are a notation device for cost accounting; standard autograd need not materialize $J_s$ or $B_s$ explicitly. With this notation, full-gradient training through a weight-tied trajectory has

$$C_{\text{full}}(T) = c_\ell + Tc_B + (T-1)c_J + (T-1)c_\theta + c_u, \tag{13}$$
$$M_{\text{full}}(T) = Ta_f + a_\ell + P. \tag{14}$$

Detached carry keeps the forward trajectory long but removes the temporal state-backward chain. For terminal-loss training with detached carry, only the final local transition contributes to the backward graph:

$$C_{\text{det}}(T) = c_\ell + c_B + c_u, \tag{15}$$
$$M_{\text{det}}(T) = a_f^{\text{det}} + a_\ell + P. \tag{16}$$

SOT then changes the optimizer interval itself from a full $T$-step trajectory to one outer-loop step:

$$C_{\text{SOT}} = c_\ell + c_B + c_u, \qquad\qquad M_{\text{SOT}} = a_f^{\text{det}} + a_\ell + P, \tag{17}$$
$$C_{\text{SOT+trunc}} = c_\ell + c_B^{\text{trunc}} + c_u, \qquad\qquad M_{\text{SOT+trunc}} = a_f^{\text{trunc}} + a_\ell + P. \tag{18}$$

*Table 9.* **Symbolic training-cost comparison across the training-dynamics variants.** Costs and retained memory are reported per optimizer interval: offline variants use the full trajectory shown in the row, whereas SOT variants update after each outer-loop step.

| Base/Variants | Fwd / interval | Bwd+Update / interval | Memory / interval |
|---|---|---|---|
| ① vanilla | $LF$ | $c_\ell + Lc_B + (L-1)c_J + Lc_u$ | $La_f + a_\ell + LP$ |
| ② weight-tied | $LF$ | $c_\ell + Lc_B + (L-1)c_J + (L-1)c_\theta + c_u$ | $La_f + a_\ell + P$ |
| ③ + full grad, $D = 2L$ | $2LF$ | $c_\ell + 2Lc_B + (2L-1)c_J + (2L-1)c_\theta + c_u$ | $2La_f + a_\ell + P$ |
| ④ + full grad, $D = 16L$ | $16LF$ | $c_\ell + 16Lc_B + (16L-1)c_J + (16L-1)c_\theta + c_u$ | $16La_f + a_\ell + P$ |
| ⑤ + detached carry, $D = 16L$ | $16LF$ | $c_\ell + c_B + c_u$ | $a_f^{\text{det}} + a_\ell + P$ |
| ⑥ + traj. sup., $D = 16L$ | $16LF$ | $16L(c_\ell + c_B) + (16L-1)c_\theta + c_u$ | $16L(a_f^{\text{det}} + a_\ell) + P$ |
| ⑦ + SOT | $F$ | $c_\ell + c_B + c_u$ | $a_f^{\text{det}} + a_\ell + P$ |
| ⑧ + trunc. | $F$ | $c_\ell + c_B^{\text{trunc}} + c_u$ | $a_f^{\text{trunc}} + a_\ell + P$ |

*Notes.* The local symbols follow the surrounding text and the derivation in Huang (2026). Only the final local transition contributes to the backward graph in the terminal-loss detached-carry row. The trajectory-supervision row uses the offline end-of-trajectory loss accumulation contract. Depth-independent constants outside the recurrent chain are omitted; a fixed $16L$-step training horizon is obtained by multiplying by the number of intervals needed to cover $16L$ outer-loop steps.

## A.3. Additional Experiments on Randomized State Initialization

This section provides supporting ablations for randomized state initialization (RI) in Sec. 5.1. The main text uses simple zero-mean Gaussian initial states to improve coverage over attractor basins under depth–breadth scaling. Here we ask two narrower questions: whether replacing this simple stochastic prior with an input-conditioned learnable initializer is beneficial, and how sensitive the method is to the Gaussian noise scale. The results support the main-text design choice: simple randomized initialization is effective, while the tested learnable initializer and scale tuning do not change the central conclusion.

**Learnable initial state.** A body of work on diffusion models suggests that the choice of initialization can significantly affect generation quality, and that learning a better initialization than standard Gaussian noise can be beneficial. For example, Zhou et al. (2025) proposes *golden noise*, where an auxiliary network learns to transform Gaussian noise into a

prompt-conditioned initialization that improves alignment. Related approaches include directly optimizing the initial noise at inference time (Guo et al., 2024), as well as reducing the initialization gap in video diffusion models through structured initialization schemes (Wu et al., 2024). These results motivate a natural question in our setting: whether learning the initial latent state $\mathbf{z}_0$ can similarly improve iterative reasoning models.

Instead of sampling $\mathbf{z}_0 \sim \mathcal{N}(0, \sigma_0 I)$, we consider a simple learnable initialization scheme. Specifically, we introduce an input-conditioned 2-layer MLP $g_\phi$ that predicts the initial state $\mathbf{z}_0 = g_\phi(\mathbf{x})$, and train $\phi$ jointly with the rest of the model parameters under the same training objective. This can be viewed as learning a conditional prior over the initial latent state.

**Empirical observation.**    In our experiments, learning the initialization does not improve the training or held-out evaluation accuracy curves under the protocol used here. Tab. 10 reports the held-out exact accuracy of the TRM baseline, TRM with randomized state initialization (TRM + RI), and TRM with a learnable initializer $\mathbf{z}_0 = g_\phi(\mathbf{x})$. The learnable initializer reaches 83.99% exact accuracy at 50k training steps, compared with 86.03% for TRM + RI and 84.06% for the TRM baseline. Thus, the tested input-conditioned initializer does not improve over simple randomized initialization, and it also does not provide a reliable gain over the baseline TRM.

*Table 10.* **Learnable initialization does not improve held-out exact accuracy.** We report exact accuracy (%) at selected training checkpoints for the learnable-initialization ablation. The best column reports the best held-out checkpoint up to 50k training steps.

| Model | 30k | 40k | 50k | Best $\leq$50k |
|---|---|---|---|---|
| TRM | 77.92 | 82.65 | 84.06 | 84.06 |
| TRM + RI | 80.43 | 84.20 | 86.03 | 86.03 |
| TRM + learnable $\mathbf{z}_0$ | 78.46 | 82.59 | 83.99 | 83.99 |

Across the full set of evaluated checkpoints up to 50k training steps, the learnable initializer never exceeds TRM + RI on held-out exact accuracy, and its best checkpoint remains slightly below the 50k result of the TRM baseline.

**Scope and limitations.**    Since we do not conduct further analysis or ablations on alternative initialization parameterizations, architecture designs, objectives, or regularization strategies, we do not make stronger claims about the general effectiveness of learnable initializations in iterative reasoning models. A more systematic study of initialization priors in latent state space is left for future work.

**Randomness scale.**    In the main text, we instantiate RI with zero-mean Gaussian initial states. This section examines how the scale of this initialization randomness affects TRM performance.

TRM maintains two latent spaces, a high-level latent $\mathbf{z}_H$ and a low-level latent $\mathbf{z}_L$. We therefore vary the noise scale for each latent separately by sampling $\mathbf{z}_H \sim \mathcal{N}(0, \sigma_H I)$ and $\mathbf{z}_L \sim \mathcal{N}(0, \sigma_L I)$. When a noise scale is not explicitly specified, we use the default $\sigma = 1$. Setting $\sigma = 0$ corresponds to a deterministic (fixed) initialization.

Overall, randomized state initialization can improve performance over the deterministic fixed-initialization run in this sweep, and the choice of noise scale matters. For example, randomizing only $\mathbf{z}_H$ (setting $\sigma_H = 1, \sigma_L = 0$) improves exact accuracy from 84.06% to 86.29%, and randomizing only $\mathbf{z}_L$ (setting $\sigma_H = 0, \sigma_L = 1$) improves it to 86.25%. For the joint sweep, the evaluated settings with $\sigma_H = 1$ achieve exact accuracies 86.03%, 86.83%, 87.30%, and 86.85% for $\sigma_L \in \{1, 4, 8, 16\}$, respectively. Fixing $\sigma_L = 1$ and increasing $\sigma_H$ gives 86.03%, 86.38%, 86.08%, and 86.29% for $\sigma_H \in \{1, 4, 8, 16\}$. Among the tested settings, the best performance is achieved by combining moderate noise on $\mathbf{z}_H$ with a larger noise scale on $\mathbf{z}_L$, peaking at $\sigma_H = 1, \sigma_L = 8$ with exact accuracy 87.30%. These observations are consistent with the view that initialization randomness is most helpful when it places the iterative dynamics within basins that lead to correct solutions, whereas overly large perturbations may not further improve (and can slightly reduce) accuracy.

Due to limited computational resources, we report results from a single run for each configuration and do not include variance estimates across multiple random seeds. A more systematic study of variability across runs and a denser sweep of noise scales are left for future work.

### A.4. Additional Experiments on Path Stochasticity

As shown in Eq. 2, we introduce path stochasticity by injecting step-wise noise into the iterative update, $\mathbf{z}_{k+1} = \mathbf{z}_k + (1 - \lambda) \, r_\theta(\mathbf{z}_k; \mathbf{x}) + \beta \, \varepsilon_k$, where $\varepsilon_k \sim \mathcal{N}(0, I)$ by default.

*Table 11.* **Path independence of different methods.** We report $\Delta_{\mathrm{PI}}(B{=}128)$ at $D{=}16$ on Sudoku-Lite and Maze-Unique; lower is better.

| Method | $\Delta_{\mathrm{PI}}$ (%)↓ Sudoku | $\Delta_{\mathrm{PI}}$ (%)↓ Maze |
|---|---|---|
| TRM | 3.58 | 28.60 |
| + RI | 0.10 | 3.73 |
| + RI + NI | 0.13 | 1.33 |

*Table 12.* **Noise-scale ablation.** Accuracy (%) for fixed and learned noise.

| Method | Eval | Train |
|---|---|---|
| $\beta{=}0.01$ | 86.4 | 84.7 |
| $\beta{=}0.05$ | 85.9 | 87.2 |
| $\beta{=}0.1$ | 86.3 | 85.1 |
| Learned $\epsilon$ | 87.1 | 85.3 |

Tab. 12 reports a brief ablation over the fixed Gaussian noise scale $\beta$, along with a learned-noise variant. Moderate Gaussian noise yields comparable performance to the default setting, whereas overly large noise can slightly reduce evaluation accuracy. Although the learned-noise variant performs best in this small ablation, Tab. 13 shows that it does not consistently improve when the number of stochastic samples or restarts $S$ is increased. We therefore use fixed Gaussian noise by default and keep learned noise as an ablation rather than adding extra noise parameters to the main method.

*Table 13.* Exact accuracy (%) as a function of the number of stochastic restarts $S$. Increasing $S$ improves performance for both settings, with learned noise helping at smaller budgets and fixed $\beta{=}0.01$ slightly overtaking it at the largest sampling budgets.

| Setting | S16 | S32 | S64 | S128 | S256 | S512 | S1024 | S2048 |
|---|---|---|---|---|---|---|---|---|
| Learned $\epsilon$ | 85.99 | 89.99 | 92.14 | 93.75 | 94.63 | 95.61 | 95.85 | 95.95 |
| $\beta = 0.01$ | 84.28 | 87.35 | 90.77 | 92.97 | 94.63 | 95.07 | 96.09 | 96.78 |

## A.5. Generalization Beyond the Main Setting

We further test whether the proposed training-and-scaling recipe is tied to the main Sudoku-Extreme and Maze-Unique settings. Tab. 14 reports two complementary checks: performance on Mini-ARC (Kim et al., 2022), and transfer from the Sudoku MLP-token-mixer backbone to a self-attention Transformer backbone.

*Table 14.* Generalization beyond the main experimental setting. EqR improves over baselines on Mini-ARC, and the same recipe also transfers from the MLP-token-mixer backbone to a self-attention Transformer backbone.

| Setting | Method | Accuracy (%) |
|---|---|---|
| Mini-ARC | HRM | 44.85 |
| Mini-ARC | TRM | 48.35 |
| Mini-ARC | EqR | **55.28** |
| Sudoku-Extreme (MLP-mixer) | Baseline | 84.1 |
| Sudoku-Extreme (MLP-mixer) | + training intervention | 86.4 |
| Sudoku-Extreme (MLP-mixer) | + inference scaling | **99.8** |
| Sudoku-Extreme (Transformer) | Baseline | 72.0 |
| Sudoku-Extreme (Transformer) | + training intervention | 74.7 |
| Sudoku-Extreme (Transformer) | + inference scaling | **95.9** |

On Mini-ARC, EqR reaches $55.28\%$ exact accuracy, improving over both HRM ($44.85\%$) and TRM ($48.35\%$). On Sudoku-Extreme, the same pattern appears across two token mixers. For the MLP-token-mixer backbone used in the main Sudoku setting, the training interventions improve accuracy from $84.1\%$ to $86.4\%$, and inference scaling raises it further to $99.8\%$. For the Transformer backbone, the corresponding numbers are $72.0\%$, $74.7\%$, and $95.9\%$. These results suggest that the gains are not limited to the main Sudoku-Extreme and Maze-Unique experiments, nor to the specific MLP-token-mixer backbone used in the standard Sudoku-Extreme setup.

## A.6. Seed Stability Diagnostics

We check whether the same-budget gain is stable across independent random seeds for EqR. Across five seeds at 50k training steps, the baseline reaches $84.33 \pm 0.59\%$ exact accuracy (95% CI: [83.59, 85.07]), whereas EqR reaches $86.18 \pm 0.44\%$ (95% CI: [85.63, 86.72]). Thus, EqR remains higher than the baseline at the same budget and shows slightly lower seed-to-seed variation.

## B. Qualitative Study

Fig. 13 visualizes one TRM reasoning trajectory on a Sudoku-Extreme puzzle. Each panel decodes the model's current latent state into a full Sudoku grid after one iteration step. Early iterations already propose many correct entries, but later iterations continue to revise both correct and incorrect cells. The circled cell illustrates this non-monotonic behavior: the decoded value alternates across several candidates before the trajectory eventually settles into a consistent solution.

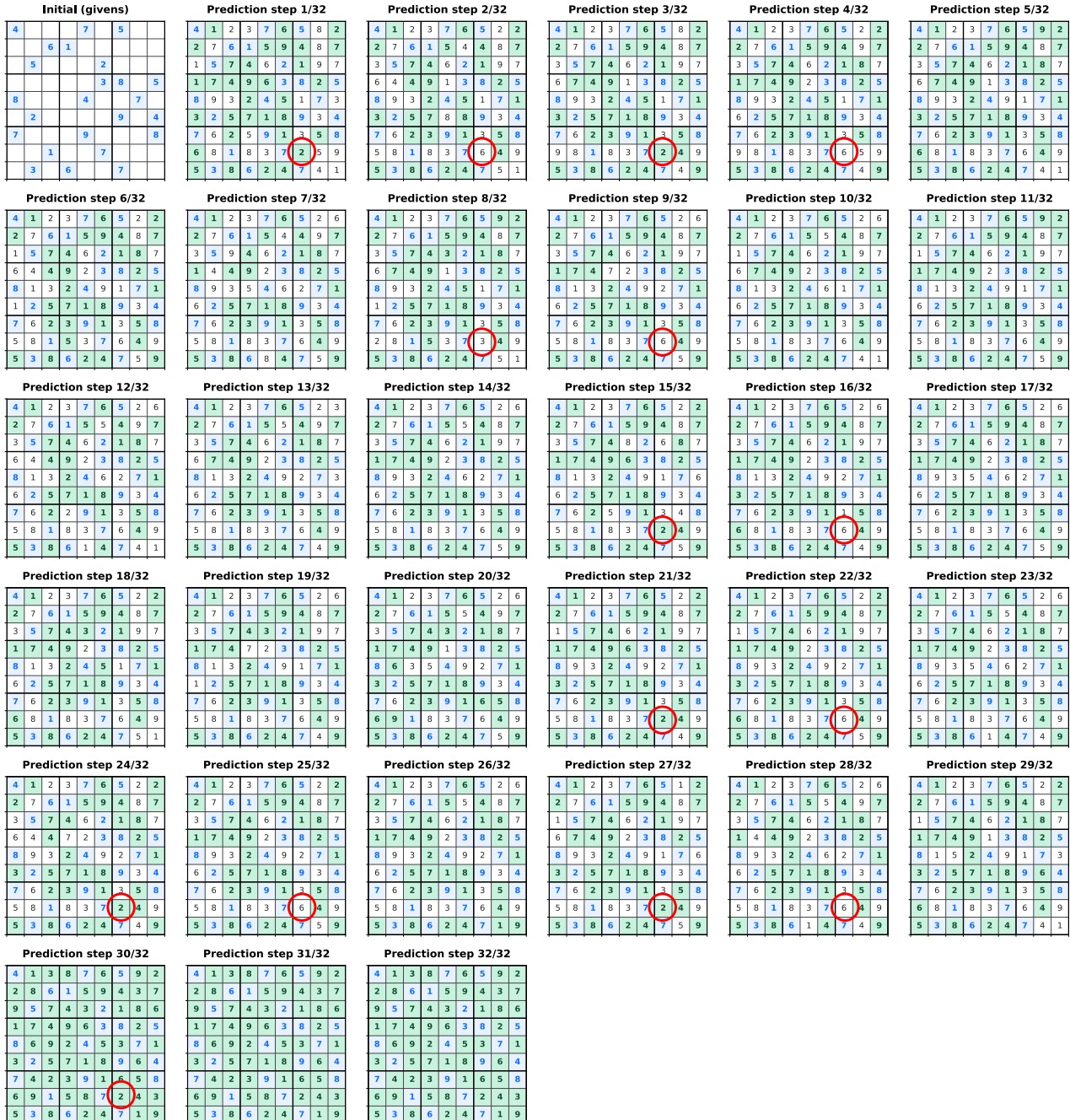

*Figure 13.* An example reasoning trajectory (length 32) produced by TRM on a Sudoku-Extreme puzzle. Correct predictions are highlighted in green. The figure suggests that the model's reasoning is not strictly sequential in the way an algorithmic solver would be. Instead, it exhibits "erase then retry" behavior, with partial revisions that overwrite earlier choices. We mark one representative location with a red circle, where the model oscillates between 2 and 6, with 3 appearing once at step 8.

# C. Dataset Details and Task Definitions

This appendix defines the dataset variants and task conventions used throughout our experiments, with particular attention to the distinction between the original Maze-1k benchmark and our uniquely solvable Maze-Unique setting.

## C.1. Dataset and Benchmark Specifications

**Sudoku-Extreme** Sudoku-Extreme (Wang et al., 2025) is a benchmark of exceptionally challenging $9 \times 9$ Sudoku instances designed to stress long-horizon constraint satisfaction. According to the HRM paper, it remains difficult even for strong modern reasoning models such as DeepSeek-R1 and Claude 3.7 8k. We reuse the code and data released with the Sudoku-Extreme dataset from HRM (Wang et al., 2025).

**Sudoku-Lite.** As shown in Tab. 16a, the validation set of Sudoku-Extreme is very large. To improve evaluation efficiency, we introduce a subset of 2048 cases sampled uniformly at random from Sudoku-Extreme, which we term Sudoku-Lite.

Tab. 15 compares exact accuracy on Sudoku-Extreme and Sudoku-Lite. Across multiple model variants, performance on Sudoku-Lite is slightly worse, suggesting that this smaller evaluation subset is not an easier benchmark for current models.

| Dataset | TRM | HRM | EqR |
|---|---|---|---|
| Sudoku-Extreme | 84.1 | 61.0 | 86.4 |
| Sudoku-Lite | 82.0 | 57.8 | 84.3 |

*Table 15.* Exact accuracy (%) on Sudoku-Extreme and Sudoku-Lite.

**Maze-1k.** Maze-1k evaluates shortest-path prediction on a $30 \times 30$ grid with obstacle cells. The benchmark follows the instance-generation procedure of Lehnert et al. (2024), as used by HRM. Many instances admit multiple shortest paths, while the released dataset provides one labeled path per input; Fig. 14a shows a representative example.

**Maze-Unique.** We construct Maze-Unique as a controlled $30 \times 30$ variant in which every retained instance has a unique shortest path. Each maze is generated as a perfect maze, i.e., a tree-structured grid with a single simple path between any pair of cells. For each maze, we repeatedly sample start-goal pairs, compute the shortest-path length, retain pairs within the target length range, and de-duplicate mazes across the training and test splits. The final dataset contains 1,000 training instances and 1,000 test instances. As shown in Tab. 16b, this filtering does not make the task shorter by path length: Maze-Unique has a slightly larger average shortest-path length than Maze-1k. A concrete example is shown in Fig. 14b.

| Task | Board Size | Vocab Size | Train Size | Val Size |
|---|---|---|---|---|
| Maze-Unique | $30 \times 30$ | 6 | 1000 | 1000 |
| Sudoku-Extreme | $9 \times 9$ | 11 | 1,001,000 | 422,786 |

| Dataset | Avg. Path Length |
|---|---|
| Maze-1k | 113.79 |
| Maze-Unique | **119.75** |

*(a)* Dataset specifications.

*(b)* Maze path-length statistics.

*Table 16.* **Dataset statistics.** In (a), we report the board size, vocabulary size, and split sizes used in our experiments. Following Wang et al. (2025), we augment the Sudoku-Extreme training set 1000-fold, yielding $1000 \times (1000 + 1)$ examples. In (b), Maze-Unique is not obviously easier by path length alone: its average shortest-path length is slightly larger than Maze-1k.

**Naming convention.** In the main text, for simplicity, *Sudoku* denotes Sudoku-Extreme and *Maze* denotes Maze-Unique, matching the shorthand introduced in Sec. 6. Sudoku-Extreme is the full HRM benchmark used for the main Sudoku results, Sudoku-Lite is our 2048-example evaluation subset sampled from Sudoku-Extreme, Maze-1k is the original ambiguous shortest-path benchmark used by HRM/TRM, and Maze-Unique is our uniquely solvable maze benchmark used for the main Maze results.

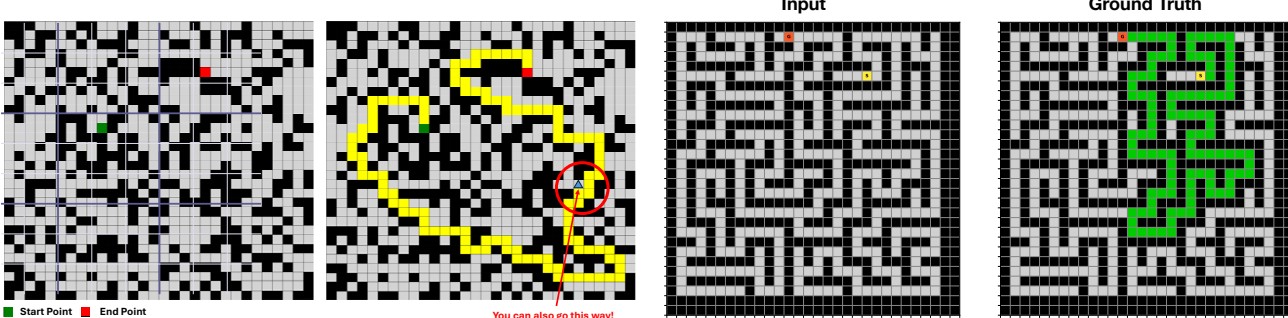

*(a)* Maze-1k instance from the HRM paper (Fig. 6c); the shortest path is not unique.

*(b)* Maze-Unique instance with a unique shortest path between the start and goal.

*Figure 14.* **Maze benchmark examples.** Maze-1k contains inputs with multiple valid shortest paths, while Maze-Unique restricts the benchmark to uniquely solvable instances.

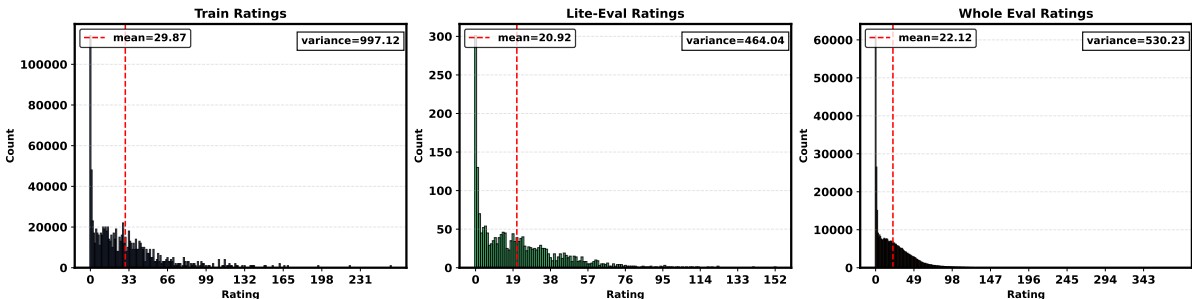

*Figure 15.* **Distribution of Sudoku-Extreme difficulty ratings.** Ratings are derived from the backtrace length of a code-based solver and are provided by the HRM dataset release (https://huggingface.co/datasets/sapientinc/sudoku-extreme).

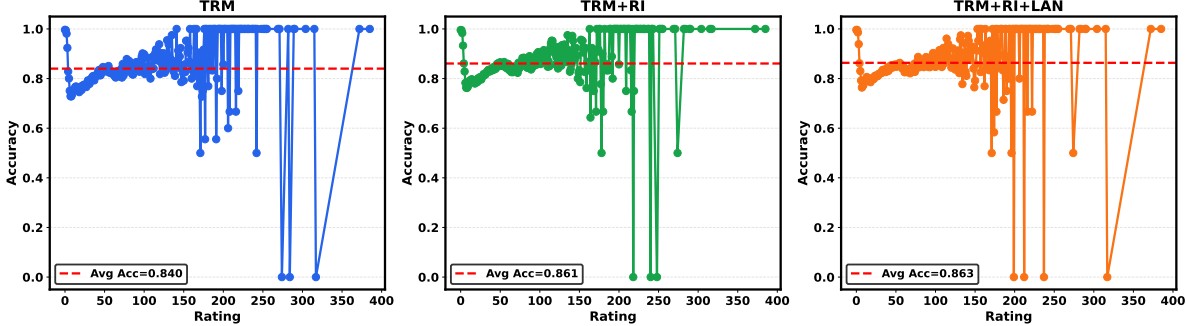

*Figure 16.* **Model accuracy versus Sudoku-Extreme difficulty rating.** The model solves some extremely difficult instances while failing on some easier puzzles, suggesting that solver-based difficulty ratings do not fully predict model success.

## C.2. Datasets and Task Definitions Shape Attractor Landscapes

We find that when the dataset is ill-defined with respect to the training target, attractor dynamics cannot be learned reliably. On the original Maze-1k dataset used in the HRM and TRM papers, iterative models fail to exhibit stable test-time scaling, and training remains unstable even after extensive hyperparameter tuning.

The root cause is label ambiguity rather than model capacity or optimization. The Maze-1k task is defined as finding a shortest path from start to goal, yet for most of the mazes in this dataset the shortest path is not unique (see Fig. 14a). Despite this, the dataset provides only a single target trajectory per maze, effectively casting a one-to-many task as a one-to-one supervised learning problem. According to the HRM paper, this dataset is generated using the codebase of Lehnert et al. (2024). That work explicitly distinguishes two variants of maze-style planning data. In the deterministic variant, the search procedure is fixed and produces a unique trace and solution for each maze. In the non-deterministic variant, randomized tie-breaking during search yields multiple equally valid shortest paths for the same maze input. For the non-deterministic variant, Lehnert et al. (2024) therefore report any-correct or any-optimal metrics, acknowledging that multiple outputs should be considered correct.

In contrast, HRM and TRM are trained on non-unique Maze-1k data using token-level cross entropy against a single provided trajectory, creating a mismatch between the training objective and the task. From a landscape perspective, the task admits multiple correct attractors, yet the loss artificially designates one arbitrary attractor as the sole target and penalizes all others. Consequently, the learned attractor landscape is misaligned with correctness: nearby trajectories may converge to alternative valid paths that nonetheless incur non-negligible loss, yielding multiple shallow and competing attractors. This destroys stable depth scaling, and randomized state initialization or noise injection becomes counterproductive, as improving coverage or stability of one attractor can degrade performance when there are actually multiple valid attractors.

To isolate this effect, we construct Maze-Unique, where each maze admits a unique shortest path. Under this setting, supervision aligns with the task structure, and iterative models recover stable attractor dynamics and meaningful test-time scaling behavior. When scaling up depth at test time, we found that models trained on Maze-Unique improve as NFE increases, whereas models trained on the original Maze-1k dataset stay flat or slightly degrade. While direct metric comparisons across datasets are not meaningful due to differences in model size and setup, the qualitative difference in stability and scaling behavior is clear.

The dataset fundamentally determines the geometry of both the optimization objective and the learned attractor landscape. When a task admits multiple valid solutions, imposing single-solution supervision makes learning ill-posed. This misspecification prevents stable attractor landscapes from forming and, as a result, breaks test-time scaling.

---

**Algorithm 1** Pseudocode with RI randomized state initialization and **NI** noise injection.

---

```
# Given: x, n, T, λ, β
# x: input condition
# n: inner loop updates for z_L per outer step
# T: number of outer steps
# λ: damping coefficient
# β: noise injection strength
# σ_H, σ_L: RI init std for z_H, z_L
# N_sup: max outer supervision steps

def iter_step(z, cond):
    ε = noise() # NI path stochasticity
    return z + (1 - λ) * (f_θ(z, cond) - z) + β * ε

def latent_loop(z_H, z_L):
    for _ in range(n):
        z_L = iter_step(z_L, x + z_H)
    z_H = iter_step(z_H, z_L)
    return z_H, z_L

def truncated_unroll(z_H, z_L):
    # detached carry for grad truncation
    with no_grad():
        for _ in range(T - 1):
            z_H, z_L = latent_loop(z_H, z_L)

    # gradient tracked final outer step
    z_H, z_L = latent_loop(z_H, z_L)

    ŷ, q̂ = lm_head(z_H), q_head(z_H)
    return ŷ, q̂, z_H, z_L

# RI randomized state initialization
z_H ~ 𝒩(0, σ_H I);  z_L ~ 𝒩(0, σ_L I)

for k in range(N_sup):
    # segmented online training update
    (ŷ, q̂, z_H, z_L) = truncated_unroll(z_H, z_L)
    L = CE(ŷ, gt) + BCE(q̂, 1[ŷ = gt])
    backprop(L); opt_step()
    if q̂ > 0: # a difficulty aware buffer
        break
```

---

# D. Method and Experimental Details

This section collects the implementation, hyperparameter, baseline-tuning, and evaluation details needed to reproduce and interpret the experimental results.

## D.1. Architecture and Hyperparameters

Here we introduce the hyperparameters used in the experiments. For Sudoku-Extreme, we follow the architecture used in the TRM paper, keeping the model components and training procedure identical unless otherwise specified.

For Maze-Unique, we deliberately reduce model capacity by decreasing the number of layers from 2 to 1 and the hidden dimension from 512 to 128. We find that larger models can reach near-perfect performance on this task, which makes the effects of different test-time-scaling variants difficult to observe. By constraining model capacity, we encourage the model to rely on iterative refinement, making the attractor dynamics easier to study. Detailed configurations are shown in Tab. 17.

*Table 17.* Hyper-parameters for TRM, HRM, URM, and EqR on the Sudoku and Maze datasets. Equivalent layers are reported per outer iteration. We use the Adam-atan2 optimizer (Everett et al., 2024; Kingma & Ba, 2015), with constant learning rate $10^{-4}$, weight decay 1.0, EMA ratio 0.999, a 2k-step linear warmup, and $(\beta_1, \beta_2) = (0.9, 0.95)$. "–" indicates that the field does not apply. For each model and dataset, we report the best checkpoint observed during training for fair comparison.

| Hyper-parameter | Model | | | | | | | |
| | TRM | | HRM | | URM | | EqR | |
| | Sudoku | Maze | Sudoku | Maze | Sudoku | Maze | Sudoku | Maze |
|---|---|---|---|---|---|---|---|---|
| H-Layer | – | – | 4 | 1 | 4 | 1 | – | – |
| L-Layer | 2 | 1 | 4 | 1 | – | – | 2 | 1 |
| H-Cycle | 3 | 3 | 2 | 2 | 12 | 12 | 3 | 3 |
| L-Cycle | 6 | 4 | 2 | 2 | – | – | 6 | 4 |
| Outer Loop | 16 | 16 | 16 | 16 | 16 | 16 | 16 | 16 |
| #Equivalent Layers | 42 | 15 | 12 | 6 | 24 | 12 | 42 | 15 |
| Hidden Size | 512 | 128 | 512 | 128 | 512 | 128 | 512 | 128 |
| #Param (M) | 5.03 | 2.64 | 27.28 | 5.26 | 13.67 | 2.65 | 5.03 | 2.64 |
| Global Batch Size | 768 | 768 | 768 | 768 | 128 | 768 | 768 | 768 |
| Training Steps | 50k | 100k | 50k | 100k | 50k | 100k | 50k | 100k |

## D.2. Learning-Rate Control for Feedforward Baselines

To rule out the possibility that feedforward baselines underperform due to suboptimal learning-rate choices, we sweep three learning rates $\{5 \times 10^{-4}, 5 \times 10^{-5}, 1 \times 10^{-4}\}$ and inspect the resulting training and evaluation trajectories. Across this sweep, learning-rate tuning does not remove the feedforward generalization gap: the feedforward models can fit the training set, but evaluation accuracy remains extremely low. In particular, the 4-layer MLP moves from underfitting at the smallest learning rate to high training accuracy at larger learning rates, whereas the 16-layer MLP fits the training set across all three learning rates. This indicates that the feedforward baselines mainly memorize the Sudoku-Extreme training distribution rather than learning a structure that generalizes.

## D.3. Evaluation Metrics

This subsection collects the formal definitions of the evaluation metrics used in Sec. 6.

**Averaged exact acc.** We report *averaged exact acc.* under $B$ independent restarts:

$$\text{AccAvg}(B; \mathbf{x}) := \frac{1}{B} \sum_{i=1}^{B} \mathbf{1}\{\hat{y}^{(i)}(\mathbf{x}) = \mathbf{y}\}. \tag{19}$$

When $B = 1$, this reduces to standard single-run accuracy.

**Top-1 convergence accuracy.** Given $B$ independent restarts and $T$ iteration steps, we select the trajectory whose final states exhibit the strongest convergence, measured by the mean residual over the last $L$ iterations,

$$r_{T,L}^{(i)}(\mathbf{x}) = \frac{1}{L} \sum_{t=T-L+1}^{T} \big\| f_\theta(\mathbf{z}_t^{(i)}; \mathbf{x}) - \mathbf{z}_t^{(i)} \big\|.$$

We then report whether the corresponding prediction is correct:

$$\mathrm{Top1Conv}(B; \mathbf{x}) \ := \ \mathbf{1}\Big\{ \hat{\mathbf{y}}^{(i^\star)}(\mathbf{x}) = \mathbf{y} \Big\}, \tag{20}$$

where $i^\star \in \arg\min_{i \in \{1,\dots,B\}} r_{T,L}^{(i)}(\mathbf{x})$. Unless otherwise stated, we use a convergence window of $L = 3$ iterations.

**Majority vote accuracy.** As a complementary baseline under breadth scaling, we report majority vote accuracy over the $B$ independent restarts,

$$\mathrm{MajVote}(B; \mathbf{x}) \ := \ \mathbf{1}\Big\{ \mathrm{mode}\big(\{\hat{\mathbf{y}}^{(i)}(\mathbf{x})\}_{i=1}^{B}\big) = \mathbf{y} \Big\}. \tag{21}$$

**Path independence across restarts.** We quantify inference stability by measuring how sensitive accuracy is to restart randomness. For an input $\mathbf{x}$, let $\bar{\mathrm{Acc}}_B(\mathbf{x})$ denote the mean exact accuracy over $B$ independent restarts. We define

$$\Delta_{\mathrm{PI}}(B) \ := \ \mathbb{E}_{\mathbf{x}}\big[ \big| \bar{\mathrm{Acc}}_B(\mathbf{x}) - \bar{\mathrm{Acc}}_1(\mathbf{x}) \big| \big]. \tag{22}$$

Smaller $\Delta_{\mathrm{PI}}(B)$ indicates stronger path independence.

### D.4. Compute Accounting for Iterative Inference

We count iterations at the outer-loop level unless noted otherwise. We use $D$ for the number of outer iterations in one trajectory and $B$ for the number of independent restarts. We use the number of function evaluations, $\mathrm{NFE} = D \cdot B$, to describe inference budgets; each function evaluation corresponds to one outer-loop iteration.

One outer-loop iteration can contain several real layer applications. For Sudoku-Extreme, the update function applies two real layers over 21 inner loops, so one outer iteration corresponds to $N_{\mathrm{eq}} = 2 \cdot 21 = 42$ equivalent layers. A single trajectory with $D = 1024$ therefore reaches $D \cdot N_{\mathrm{eq}} = 1024 \cdot 42 = 43{,}008$ equivalent layer evaluations. This is the source of the "over 40,000 layers" statement in the main text.

Breadth scaling multiplies the total budget, but it does not change the depth of any individual trajectory. For total equivalent-layer accounting, $\mathrm{NLE} = D \cdot B \cdot N_{\mathrm{eq}}$. The best Sudoku-Extreme result in Tab. 4 uses $D = 64$ and $B = 128$, i.e., 8192 function evaluations and $64 \cdot 128 \cdot 42 = 344{,}064$ total equivalent layer evaluations. We describe this setting as two-axis scaled inference rather than a 344k-depth unroll, since each trajectory still has depth $D = 64$.

## E. Extended Related Work and Discussion

This appendix expands the related work discussion in the main text.

**Deep Equilibrium Models.** Our framework is closely related to implicit models, especially Deep Equilibrium Models (DEQs) (Bai et al., 2019). Related work extends this implicit view through path-independent equilibria for exploiting test-time computation (Anil et al., 2022) and practical training methods for implicit models (Geng et al., 2021a;b; Fung et al., 2022; Gu et al., 2020). Rather than stacking a fixed number of explicit layers, a DEQ defines the representation as the fixed point of a weight-tied nonlinear transformation. Concretely, a DEQ solves for an equilibrium $\mathbf{z}^\star$ satisfying $\mathbf{z}^\star = f_\theta(\mathbf{z}^\star; \mathbf{x})$. Under convergence, this can be viewed as the implicit limit of an infinitely deep weight-tied network.

We borrow the same fixed-point vocabulary, but study a different question. DEQ work primarily uses convergence to an equilibrium as a representation-learning and training device; our goal is to understand when the learned latent space dynamics make convergence reliable for solving a task. In our setting, reaching some fixed point is not enough: the model must shape a landscape whose large, stable basins correspond to correct solutions rather than spurious or unstable attractors. Thus attractor coverage, stability, and basin size provide a language for explaining why iterative reasoning succeeds, fails, or benefits from additional inference-time updates.

**Latent reasoning models.** We use *latent reasoning models* to refer to methods that spend additional computation in latent space before emitting externally visible tokens or final predictions. This latent computation can be organized along two axes. The vertical axis increases computation by repeatedly updating a latent state with weight-tied iterations, as in weight-tied language models (Geiping et al., 2025; Zhu et al., 2025), PonderLM (Zeng et al., 2026), Parcae (Prairie et al., 2026), LoopViT (Shu et al., 2026), and the HRM/TRM/URM line of structured reasoners (Wang et al., 2025; Jolicoeur-Martineau, 2025; Gao et al., 2025). The horizontal axis inserts additional latent positions before a visible output: pause tokens delay answer extraction while the decoder processes extra hidden vectors (Goyal et al., 2024), Coconut feeds the last hidden state back as a continuous thought (Hao et al., 2025), SoftCoT and SoftCoT++ construct soft thought tokens for efficient or diverse continuous-space reasoning (Xu et al., 2025a;b), and PonderLM-2 pretrains latent thoughts before actual token prediction (Zeng et al., 2025). Recent implicit-CoT variants further supervise, align, or score these latent tokens to reduce collapse and improve interpretability or parallel test-time scaling (Wei et al., 2026; Chu et al., 2026; You et al., 2025). These horizontal methods internalize or compress chain-of-thought-like computation into latent tokens, whereas our work focuses on the vertical fixed-state dynamics of iterative reasoners: we ask when extra updates help by studying whether training shapes a solution-aligned attractor landscape, rather than treating iteration count or hierarchy alone as the source of generalization.

**Theoretical and mechanistic analyses of weight-tied iteration.** A useful way to organize this line of work is as a progression from what weight-tied iteration can compute, to what one more iteration is worth, and finally to what hidden-state mechanisms make iteration succeed or fail. The first question is computational: programmable-computer constructions show that a shallow weight-tied Transformer can emulate counters, branches, function calls, and small instruction-set programs when the sequence supplies both instructions and memory (Giannou et al., 2023). Learning-algorithm experiments show that adding iteration lets a Transformer fit data with accuracy comparable to a much larger standard Transformer (Yang et al., 2024). Expressivity analysis then identifies limits that are specific to repeated shared blocks and shows that timestep encodings recover additional iteration-dependent behavior (Xu & Sato, 2025). Latent-thought theory makes the same point from the reasoning side: on synthetic tasks where effective depth is the scarce resource, iterating a small block can approximate a much deeper non-weight-tied model (Saunshi et al., 2025). Together, these results justify treating iterations as real computation, but they do not yet say which trajectories through state space are reliable.

The second question is resource accounting: how much does an iteration buy, and what state must be preserved across iterations? Iso-depth scaling estimates that one extra iteration is only partially equivalent to adding a fresh layer (Schwethelm et al., 2026), while inverse-depth scaling argues that ordinary LLM depth can look like many similar layers averaging errors rather than composing qualitatively different transformations (Liu et al., 2026). Circuit-complexity analyses make the analogous point for horizontal token-space computation: intermediate decoding steps can serve as recurrent state, giving decoder-only Transformers more formal power as the chain-of-thought budget grows (Merrill & Sabharwal, 2024). This line makes iteration a measurable compute resource, but it still mostly reasons about capacity, scaling, memory, or stopping rules instead of the geometry of correct and incorrect solution states.

The third question is mechanistic: what structure appears in the latent states themselves? Controlled implicit-reasoning experiments show systematic generalization, depth extrapolation, and overthinking as the number of iterations changes (Kohli et al., 2026). Mechanistic analyses of weight-tied reasoning language models find cyclic trajectories approaching distinct fixed points, with attention behavior stabilizing across iterations (Blayney et al., 2026). Preference probes show that pairwise differences between iteration states can encode relational signals (Kirin, 2026), while latent-chain-of-thought probes caution that hidden states need not be directly readable as natural-language reasoning traces (Lu et al., 2025). Closest to our vocabulary, fixed-point analyses characterize stability through reachability, input-dependence, and geometry (Labovich, 2026), and two-scale trajectory studies distinguish small within-block refinements from larger cross-block drift (Pappone et al., 2025). Concurrent HRM analysis also links failures to fixed-point violations and multiple fixed points, but its strongest Sudoku gains rely on data augmentation, input perturbation, and model bootstrapping (Ren & Liu, 2026). Our work takes the next step: we directly measure the attractor landscape of finite-iteration, fixed-state weight-tied models, relate residual-state updates to iteration steps, and explain failure modes as trajectories entering unstable or wrong basins. We then use this landscape view to ask which training interventions reshape the basins, why those changes raise the inference-time upper bound, and how learned stopping can save compute without discarding the iterations that actually move a trajectory toward a correct attractor.

**Geometry of learning and flat minima.** The connection between the geometry of the loss landscape and a model's generalization ability is a long-standing area of research. A central hypothesis, dating back to the work of Hochreiter &

Schmidhuber (1997), posits that optimizers that converge to "flat" minima in the parameter space tend to produce models that generalize better than those that converge to "sharp" minima. The intuition is that a flat region of the loss landscape is more robust to small perturbations in the model's weights, such as those caused by shifts between the training and test data distributions. This idea has been revitalized by modern optimization methods like Sharpness-Aware Minimization (SAM) (Foret et al., 2021). SAM formalizes this intuition into a min-max optimization objective that explicitly searches for parameters residing in neighborhoods with uniformly low loss values. By doing so, SAM encourages convergence to flatter regions of the parameter landscape, leading to improvements in generalization across various benchmarks.

Our work translates this concept of robustness from parameter space to state space. While SAM seeks solutions that are stable with respect to perturbations of the model weights $\theta$, our framework seeks attractors that are stable with respect to perturbations of the latent state $\mathbf{z}$. The path stochasticity we introduce during training (section 5.2) serves a similar purpose to the perturbation step in SAM: it forces the model to learn an update function $f_\theta$ that defines a smooth and robust attractor landscape, where trajectories reliably converge to the correct solution despite noise. In essence, we are searching for "flat minima" in the landscape of the state-space dynamics, not just in the landscape of the training loss.

