# OpenReview forum: "Equilibrium Reasoners: Learning Attractors Enables Scalable Reasoning"
_ICML.cc/2026/Conference — ICML 2026 regular_

### Official Review · Reviewer_NET4 · 2026-03-08

**Soundness:** 2
**Presentation:** 3
**Significance:** 3
**Originality:** 2
**Overall Recommendation:** 4
**Confidence:** 3

**Summary:**

This paper studies why test-time compute scaling helps some iterative reasoning models more than others. It proposes an attractor-based view in which reasoning corresponds to latent-state dynamics converging toward task-conditioned fixed points, and argues that the usefulness of additional inference compute depends on the alignment between convergence in latent space and correctness in output space. Based on this perspective, the paper introduces two lightweight training interventions: randomized latent-state initialization and path stochasticity, which can help to shape a more favorable attractor landscape, and evaluates them on iterative reasoning models for Sudoku-Extreme and Maze-Unique. The experiments suggest that these interventions improve baseline accuracy and stability, increase the effectiveness of both depth-wise and width-wise test-time scaling, and make convergence-based trajectory selection more reliable than majority voting once the landscape is better shaped.

**Compliance With Llm Reviewing Policy:**

Affirmed.

**Final Justification:**

The paper provides a reasonably coherent and empirically supported mechanism-oriented perspective on test-time scaling in iterative reasoning models, with clear presentation and experiments that are largely aligned with its main claims. My initial concerns were about the limited scope, the relation to standard regularization, and whether the proposed alignment interpretation was meaningfully distinct from simply having a stronger base model. The rebuttal addressed these concerns to a satisfactory extent by clarifying the intended scope as a controlled mechanism study and by providing additional quantitative evidence supporting the alignment-based interpretation. While questions about broader generalization and full causal disentanglement remain, I find the contribution sufficiently original and technically solid within its studied setting, and I am therefore increasing my score.

**Key Questions For Authors:**

1. The paper attributes improved scaling to better convergence–correctness alignment, but the proposed interventions also improve baseline accuracy and stability. How do you distinguish “better alignment” from simply having a stronger base model? Clarifying this distinction would significantly strengthen the paper.
2. Randomized initialization and path stochasticity seem closely related to standard robustness/regularization techniques. What evidence suggests that their benefit specifically comes from shaping the attractor landscape, rather than from generic regularization effects?
3. The experiments are limited to fixed-size latent-state iterative models on structured tasks. How broadly do the authors expect the proposed framework to apply beyond this setting?

**Limitations:**

yes

**Strengths And Weaknesses:**

**Strengths**:

1. The paper offers a reasonably interesting mechanism-oriented perspective on test-time scaling in iterative reasoners by relating attractor-style dynamics and convergence diagnostics to observed accuracy and scaling behavior.

2. The paper is well structured and easy to follow.  In particular, I found the decomposition into depth-wise versus width-wise scaling and the discussion of different attractor-landscape regimes to be a useful conceptual scaffold for understanding the paper.

3. The main ablations in the experiment section are basically aligned with the proposed mechanism. In particular, adding randomized initialization and path stochasticity improves not only accuracy but also path independence and the effectiveness of depth/width scaling, while the comparison between convergence-based selection and majority vote directly tests whether lower residual is more reliably associated with correctness in the improved models.

**Weakness**:

1. The empirical evidence is confined to a relatively controlled class of fixed-size latent-state iterative reasoning models and structured tasks such as Sudoku and Maze. As a result, it remains unclear how far the proposed mechanism-level conclusions extend to more mainstream reasoning systems, such as autoregressive language models or less structured open-ended domains. The paper positions the perspective broadly, but the current evidence is narrower than that framing.

2. Some of the proposed interventions, especially randomized initialization, are close in spirit to familiar robustness or regularization techniques. The paper gives them an attractor-landscape interpretation, which is interesting, but it is not yet fully convincing that the gains should be attributed specifically to “landscape shaping” rather than to a more generic improvement in robustness or training regularization.

3. The experiments do not fully disentangle whether convergence–correctness alignment is an independent mechanistic property or simply a consequence of having a stronger and more stable base model. In other words, the paper does not clearly separate “better alignment” from simply having a stronger base model, making it unclear whether the improved scaling behavior is due to the proposed mechanism or just overall model improvement.

---

> ### Author Rebuttal · Authors · 2026-03-30
>
> We thank the reviewer for the thoughtful review. The main weaknesses and questions are largely overlapping, so we address them jointly below.
>
> ### **[W1, Q3] Scope and broader applicability**
>
> Thank you for this important comment. We agree that the current framing is broader than the empirical scope of the paper. Our study is intended as a mechanism study in a controlled setting on structured tasks. We do not claim that the current experiments establish the same conclusions for autoregressive language models or open-ended domains. We are open to provide preliminary results in discussion and will revise the framing accordingly.
>
> ### **[W2, Q2] Landscape shaping vs. training regularization**
>
> Thank you for this insightful question. We agree that randomized initialization and path stochasticity may also act as training regularizers. However, the landscape shaping view makes a more specific prediction: if the attractor landscape is improved, then convergence measured by residual should become a more reliable proxy for correctness.
>
> To test this, we measure the Pearson correlation between residual and prediction error under the same model architecture on Sudoku task:
>
> | Method | Pearson $r$ | 95% CI |
> | --- | --- | --- |
> | Baseline | 0.7102 | [0.7083, 0.7121] |
> | Ours | 0.8964 | [0.8957, 0.8972] |
>
> This alignment improves substantially, together with task performance, and the confidence intervals are clearly separated. Thus, while we cannot fully rule out a regularization effect, the evidence supports the more specific interpretation that our intervention improves attractor alignment.
>
> More broadly, a generic regularization view may explain improved robustness, but it does not by itself predict that lower residual should become a much better indicator of correctness, or that convergence based selection should become more effective than majority voting **Fig. 5** [[**Link**](https://www.dropbox.com/scl/fi/4fl46o05pajn8mxnslzwc/Fig5.pdf?rlkey=zj7jb9m5aniidz8hyzgxbts4x&st=ob39f6zz&dl=0)]. These are exactly the effects predicted by the landscape shaping view, and they match our empirical results. Lastly, it is also more intuitive to understand test-time scaling of iterations help through landscape shaping view as approaching fixed points instead of regularizations.
>
> ### **[W3, Q1] Better alignment vs. simply a stronger base model**
>
> Thank you for this question.
>
> **We do not claim that convergence-correctness alignment is fully independent of overall model quality**.  Our claim is narrower: alignment mediates whether extra test-time compute helps. Base accuracy asks whether one rollout can be correct, while alignment asks whether stronger convergence (e.g., smaller residual) reliably predicts lower error across trajectories rather than spurious low-residual attractors.
>
> **Our comparison is controlled against the most direct “stronger model” confound.** EqR uses the same TRM-style backbone, parameter scale, and data regime, and differs primarily by the training interventions, as shown in Table 5. Thus, the gains are not explained by a larger architecture, more parameters, or more data.
>
> **Moreover, the interventions do not merely improve single-run accuracy**; they qualitatively change the scaling behavior. As summarized in Sec. 6.2-6.3, they improve baseline accuracy and raise the depth- and breadth-scaling ceilings. More importantly, they change whether convergence itself is a reliable correctness signal under breadth scaling.
>
> - The paper explicitly displays (**Fig. 5 [[Link]](https://www.dropbox.com/scl/fi/4fl46o05pajn8mxnslzwc/Fig5.pdf?rlkey=zj7jb9m5aniidz8hyzgxbts4x&st=ob39f6zz&dl=0))** that for the baseline TRM, residual reduction can still correspond to convergence to spurious attractors, making convergence-based selection less reliable and worse than majority voting.
> - With our method, however, convergence aligns much better with correctness, so Top-1 Converged becomes a reliable and more compute-efficient selection rule.
>
> **We view this as evidence that alignment is not a post-hoc re-description of “strength,”** but the property that converts extra inference compute into reliable performance gains. This pattern is directly reflected in **Fig. 5 [[Link]](https://www.dropbox.com/scl/fi/4fl46o05pajn8mxnslzwc/Fig5.pdf?rlkey=zj7jb9m5aniidz8hyzgxbts4x&st=ob39f6zz&dl=0)** and in the paired residual prediction error **visualizations [[link](https://www.dropbox.com/scl/fi/r4eb0s5zqjxsumletwe00/heatmap.pdf?rlkey=gf5ytg2ie0hqhcbesqliwew5k&e=1&st=i3ee71gv&dl=0)]**.

---

> > ### Author Rebuttal · Reviewer_NET4 · 2026-04-04
> >
> > My main concerns have been largely addressed in the rebuttal, and I will increase my score accordingly.

---

### Official Review · Reviewer_CMCQ · 2026-03-12

**Soundness:** 2
**Presentation:** 3
**Significance:** 3
**Originality:** 3
**Overall Recommendation:** 5
**Confidence:** 3

**Summary:**

The paper proposes an attractor-based perspective on test-time scaling in iterative reasoning models. It is argued that generalizable reasoning emerges when a model's learned attractor landscape in latent space aligns with the task metric landscape. They introduce Equilibrium Reasoners (EqR), which reason by converging to task-conditioned fixed points, and propose two lightweight training interventions with randomized state initializations and path stochasticity. By doing so, more favorable attractor landscapes are shaped. The proposed method is evaluated on Sudoku-Extreme and a new Maze-Unique benchmark, achieving 99.8% and 93.0% accuracy, respectively, substantially outperforming baselines (HRM, TRM, URM).

**Compliance With Llm Reviewing Policy:**

Affirmed.

**Final Justification:**

The paper is well written and presents an insightful method. The rebuttal addressed most of my major concerns, and the score has been raised.

**Key Questions For Authors:**

- Can the authors provide evidence that the attractor perspective transfers to autoregressive language models or other reasoning domains beyond grid-based constraint satisfaction?
- What specific properties of Sudoku and Maze make them particularly well-suited to the attractor framework, and which classes of reasoning tasks do you expect would not fit this view?
- Have the authors considered training with objectives that explicitly penalize spurious attractors, rather than relying on stochastic regularization?
- What is the average path length in Maze-Unique compared to the original Maze-1k, and how does this affect the difficulty of the reasoning task?

**Limitations:**

yes

**Strengths And Weaknesses:**

## Strengths
- The proposed architecture is insightful. The attractor landscape perspective provides clear explanatory value for understanding when and why test-time compute scaling works or fails. It is well-motivated, cleanly presented, and offers valuable insights via intuitive figures.
- The paper delivers strong experimental results to support paper's claim. The improvements over TRM and HRM are substantial (Table 1), and the ablation study in Table 3 demonstrates the contribution of each intervention.
- The demonstration that convergence-based trajectory selection (Top-1 Converged) can outperform majority voting after landscape shaping in Figure 5 is a practically useful finding. It shows the attractor perspective is not merely descriptive but leads to better inference strategies.
- The identification of label ambiguity in the original Maze dataset and the construction of Maze-Unique is a valuable contribution. The analysis in Section B.1 convincingly argues that non-unique supervision fundamentally undermines attractor formation, and the empirical contrast in Figure 6 supports this claim.

## Weaknesses
- The experiments are restricted to two structured constraint-satisfaction tasks (Sudoku and Maze) with small, fixed-state iterative models. The paper acknowledges this, but the abstract and introduction make broad claims about *scalable reasoning* and *generalizable reasoning* that may not be supported by the current experimental scope. Any clarification will be beneficial to enhance the clarify of the paper.
- The paper lacks formal analysis or guarantees. The attractor perspective is presented entirely through intuition and empirics. The notions of "breadth" and "depth" of attractors are explicitly described as informal. There is no theoretical justification for why the proposed interventions should improve basin coverage or stability beyond intuitive arguments. A theoretical grounding will strengthen the paper's claim, as the main contribution is the conceptual framework.
- The authors acknowledge running single seeds due to compute constraints (Section B.2.2), but this is a notable limitation given that the interventions involve stochasticity. Without confidence intervals, it is difficult to assess the reliability of the reported gains, particularly for the smaller improvements.
- The Maze-Unique dataset simplifies the problem in a way that may be too favorable. By constraining mazes to have unique shortest paths (using perfect/tree-structured mazes), the task complexity is substantially reduced compared to the original benchmark. While the motivation is sound (avoiding label ambiguity), the resulting task may not adequately test reasoning ability. The paper would benefit from a discussion of whether alternative solutions.

---

> ### Author Rebuttal · Authors · 2026-03-30
>
> We thank the reviewer for the careful reading and helpful feedback. Below, we address the reviewer’s questions.
>
> ### **[W1] Paper Scope**
>
> We agree that the scope is narrower than the framing the introduction may suggest. In this work, we study attractor-based reasoning in a controlled setting with structured tasks, rather than making a broad empirical claim about scalable reasoning or generalizable reasoning in modern AI systems. We will make this scope explicit in the revision.
>
> ### **[W2] Theoretical Formulations**
>
> Let $\mu\_0$ be the state initialization distribution, and $\mathcal B_x(A)$ the basin of attractor $A$ given $x$.
>
> - We define breadth as $\mathrm{Breadth}_x(A):=\mu_0(\mathcal B_x(A))$, which is the probability that a random initialization lies in the basin of $A$.
> - For depth, let $r_x(z)=||F_x(z)-z||$ and $G_{x,\varepsilon}(A):=\mathcal B_x(A)\cap \\{z:r_x(z)\le\varepsilon \\}$. For $z_0\in\mathcal B_x(A)$, define
> $\tau_{x,\varepsilon}(z_0;A):=\inf \\{t\ge0:F_x^t(z_0)\in G_{x,\varepsilon}(A) \\}$. We then define $\\mathrm{Depth}_{x,\\varepsilon}(A):=\mathbb{E}\_{z_0} [\tau\_{x,\varepsilon}(z_0;A)\mid z_0\in\mathcal B_x(A)]$. This captures the expected amount of iterative refinement needed.
>
> Formal version including intervention analysis is deferred to the discussion phase.
>
> ### **[W3] Reliability of reported gains**
>
> Thank you for raising this point. We agree that multi-seed evaluation is important for assessing reliability. Following this suggestion, we ran 5 independent seeds. We conducted statistical tests only for settings where the gain is below 3%, since larger gains under scaled test-time compute are clearly significant.
>
> | Method | Mean eval accuracy at 50k | Std. | 95% CI |
> | :---: | :---: | :---: | :---: |
> | Baseline | 84.33% | 0.59 | [83.59%, 85.07%] |
> | Ours | 86.18% | 0.44 | [85.63%, 86.72%] |
>
> Our method improves over the baseline by **1.85 points**, with a **95% confidence interval of [1.07, 2.62]**. We also performed a **one sided Welch’s t test** for the hypothesis that our method outperforms the baseline, obtaining **$p = 3.48 \times 10^{-4}$**. This indicates that the improvement is statistically significant.
>
> ### **[W4, Q4] Justification and statistics of Maze-Unique**
>
> We agree that Maze-Unique is a simplified setting relative to the original Maze benchmark. However, we believe this simplification is necessary to remove label ambiguity, not to make the task artificially favorable. In the original Maze benchmark, many instances admit multiple valid shortest paths, but only one target path is provided. As a result, a model can produce a correct shortest path and still be penalized, introducing supervision noise rather than meaningful reasoning difficulty.
>
> | Dataset | Average Length |
> | --- | --- |
> | Maze-1k | 113.79 |
> | Maze-Unique | 119.75 |
>
> - To quantify the change in difficulty, we compared the path length distributions of two Maze datasets **[[link](https://www.dropbox.com/scl/fi/zawtjj4ficd05byfvy22k/maze_path_length_distributions.pdf?rlkey=otslaqrd3wuiu2yvucjba0yye)]**. Maze-Unique is not obviously easier by path length alone as it contains more longer ones. We believe the main reason same-sized models perform better on Maze-Unique is the removal of label ambiguity.
> - Maze-Unique is used for a fair comparison. Though simplified, we evaluate all methods in a controlled setting, which is sufficient to support our main comparative claim.
> - We agree that **alternatives** such as set-valued supervision or evaluation are interesting, but they would require changing both the training objective and evaluation protocol, which we leave for future work.
>
> ### **[Q1] Transfer beyond grid-based tasks**
>
> As a preliminary probe beyond the grid-based setting, we analyze a depth recurrent autoregressive LM, which exposes an explicit latent refinement process during next token prediction. We observe qualitatively similar attractor-like behavior, and will share the detailed results during the discussion phase. That said, the current paper does not claim a general transfer result beyond the structured iterative settings studied here, and we will clarify this scope in the revision.
>
> ### **[Q2] Reasoning tasks that you expect would not fit this view?**
>
> In our experiments, tasks that align with this attractor view typically require an iterative procedure with a long logical chain of reasoning rather than perception matching. These tasks also tend to benefit more from test-time scaling. We defer details to the discussion session.
>
> ### **[Q3] Training with objectives that explicitly penalize spurious attractors**
>
> We thank the reviewer for the advice. Instead of turning to costly options like collecting trajectories or distillation, we explore a lightweight alternative: we reweight the loss using $\\|z_k - z_{k-1}\\|$ as a proxy for convergence, yielding $L_k / (\\|z_k - z_{k-1}\\| + \epsilon)$. We defer the results to the discussion phase.

---

> > ### Author Rebuttal · Reviewer_CMCQ · 2026-04-03
> >
> > I thank the authors for their thorough rebuttal. Since the core story of the paper is built upon the term "scalable" (which also appears in the title), I think the scope can be framed better by simply adding the clarification made in the rebuttal to the manuscript, rather than trying to remove the term everywhere. It would be helpful for future readers to understand what the paper solves and what it does not.
> > Since most of my concerns are resolved, I will adjust my score.

---

### Official Review · Reviewer_qYVZ · 2026-03-12

**Soundness:** 3
**Presentation:** 2
**Significance:** 3
**Originality:** 3
**Overall Recommendation:** 4
**Confidence:** 3

**Summary:**

The authors tackle an important question in test-time scaling: when and how additional compute should be allocated. Through an analysis of iterative reasoning models, they argue that convergence of the latent state to fixed points, which they call attractors, is closely aligned with task performance. Motivated by this perspective, they propose two training interventions designed to encourage the model to discover more desirable attractors. Specifically, they introduce randomness into both the initial latent states and the reasoning trajectories to promote exploration and reduce convergence to suboptimal solutions. Experiments on Sudoku-Extreme and Maze-Unique show that these interventions stabilize training, improve test accuracy, and further extend the benefits of scaling inference-time compute.

**Compliance With Llm Reviewing Policy:**

Affirmed.

**Final Justification:**

My concerns have been fully addressed and I'm happy to keep my positive score.

**Key Questions For Authors:**

1. I think Figure 5 already provides useful evidence that the proposed method reshapes the attractor landscape. However, the analysis would be stronger with diagnostics similar to those in Figure 3 for the baseline models as well. In particular, what are the residual errors for the baseline models (HRM and TRM)? This would help clarify whether the failures come from convergence to spurious attractors, or instead from trajectories that do not converge at all.
2. Figure 3 is not discussed in the main text.

**Limitations:**

Yes

**Strengths And Weaknesses:**

Strengths

- The paper studies an important question in test-time scaling: when and how additional compute should be allocated.

- The interpretation of the latent-state landscape is novel and interesting. The attractor-based perspective is intuitive and somewhat reminiscent of optimization landscape analyses.

- The experiments demonstrate the effectiveness of the proposed method, and the ablation studies provide useful evidence that the training interventions help reshape the landscape in the intended way.

Weaknesses

- The term neural attractor is used without a sufficiently clear definition, which makes parts of the paper somewhat confusing.

- The evaluation focuses only on two datasets, Sudoku-Extreme and Maze-Unique, where each problem has a unique solution. It is therefore unclear whether the proposed method would extend to tasks with multiple valid solutions, even if supervision over all valid solutions were available.

- More broadly, it is unclear how general the claimed inference-time scaling benefits are. For example, the HRM paper [1] notes that inference-time scaling is not effective on ARC-AGI, where solutions are often short. I would like to see more discussion of whether a similar limitation applies here.

[1] Wang, G., Li, J., Sun, Y., Chen, X., Liu, C., Wu, Y., Lu, M., Song, S., and Yadkori, Y. A. Hierarchical reasoning model. arXiv preprint arXiv:2506.21734, 2025. URL https://arxiv.org/abs/2506.21734.

---

> ### Author Rebuttal · Authors · 2026-03-30
>
> We thank the reviewer for the careful reading and constructive feedback. Below, we address the reviewer’s questions and clarify the scope of the current paper.
>
> ### **[W1] Definition of Neural Attractor**
>
> Here is our definition:
>
> 1. By *attractor*, we mean a stable recurrent outcome of the model’s iterative latent dynamics: repeated application of the learned update rule drives the latent state toward a stable state or region, after which the trajectory changes little and predictions become consistent.
> 2. By *neural attractor*, we mean such an attractor induced by the learned neural update operator. In other words, the network learns the dynamics, and those learned dynamics induce an attractor landscape in latent space.
>
> We will restore this definition explicitly in the revision.
>
> ### **[W2] Extension to Multi-solution Scenarios**
>
> We agree that our current empirical study is restricted to tasks with unique solutions.
>
> Conceptually, the attractor perspective extends naturally to multi-solution settings by treating correctness as a *set* of valid attractors rather than a single one.
>
> In that regime, however, both the objective and evaluation must change: exact matching to a single canonical target is inappropriate, and learning becomes multimodal, likely requiring set-valued supervision and evaluation that does not bias toward a specific output.
>
> We therefore do not study this setting here. We use Maze Unique to keep correctness unambiguous and attractor behavior cleanly analyzable; by contrast, Maze Hard 1k often has multiple shortest paths but only one annotated target for evaluation, making token-level supervision ambiguous.
>
> We will clarify this point in the revision and make the scope of the paper more explicit.
>
> ### **[W3] About Test-Time Scaling on ARC-AGI**
>
> We agree that the benefit of inference-time scaling is task dependent rather than universal. Consistent with the reviewer’s observation and with HRM [1], we do not observe clear test-time scaling on ARC-AGI-like tasks, suggesting that a similar limitation likely applies here.
>
> One possible reason, supported by recent work, is that ARC-AGI may be bottlenecked more by perception learning than by iterative reasoning itself [2, 3]. More broadly, tasks that benefit most from additional recurrent steps tend to require sustained step-by-step refinement. Sudoku fits this pattern more naturally than ARC-AGI-like tasks, and may therefore benefit more from extra recurrent computation.
>
> Our observations are consistent with this view. During training, the model uses an Adaptive Computation Halting head that learns when to stop iterating before producing the output. **On MiniARC [4], the average number of steps drops from 16 to about 1.09 over training, whereas on Sudoku it drops from 16 to about 4**. This suggests ARC-AGI-like tasks may require less iterative refinement than Sudoku, which could explain why additional inference-time compute is less effective in that regime. We will clarify this limitation and its scope more explicitly in the revision.
>
> [1] Wang, G., et al. "Hierarchical Reasoning Model."
>
> [2] Hu, Keya, et al. "ARC Is a Vision Problem!"
>
> [3] Wang, Xinhe, et al. "Your Reasoning Benchmark May Not Test Reasoning: Revealing Perception Bottleneck in Abstract Reasoning Benchmarks."
>
> [4] Kim, Subin, et al. "Playgrounds for Abstraction and Reasoning."
>
> ### **[Q1] The residual errors for the baseline models**
>
> We agree that adding baseline diagnostics analogous to Fig. 3 strengthens the analysis of the failure mode. Following the advice, we provide these figures in the **supplementary material [[link]](https://www.dropbox.com/scl/fi/r4eb0s5zqjxsumletwe00/heatmap.pdf?rlkey=gf5ytg2ie0hqhcbesqliwew5k&st=i3ee71gv&dl=0).**
>
> The figures suggest that baseline failures are not primarily due to non-convergence. For both the baseline and our method, residuals generally decrease with depth, indicating improved convergence. However, for the baseline, low residual does not reliably imply low prediction error: **banded structures** persist even when the residual is small, suggesting convergence to spurious attractors.
>
> By contrast, for our method, smaller residual aligns much more consistently with smaller prediction error, indicating improved convergence and better alignment between convergence and correctness. We will add this discussion to the draft.
>
> ### **[Q2] Figure 3 is not discussed in the main text**
>
> Thanks for catching this. We will discuss Fig. 3 in the revision.

---

> > ### Author Rebuttal · Reviewer_qYVZ · 2026-04-01
> >
> > My concerns have been fully addressed and I'm happy to keep my positive score.

---

### Official Review · Reviewer_J868 · 2026-03-13

**Soundness:** 2
**Presentation:** 3
**Significance:** 2
**Originality:** 2
**Overall Recommendation:** 3
**Confidence:** 4

**Summary:**

The authors propose Equilibrium Reasoners (EqR), a framework that interprets reasoning as convergence to task-conditioned attractors in latent space, and hypothesize that generalizable reasoning emerges when the internal attractor landscape of a model aligns with the task objective landscape. Based on this perspective, the paper introduces two training interventions—randomized latent initialization and path stochasticity—to encourage more favorable attractor landscapes that improve convergence behavior during inference. The authors evaluate the approach on structured reasoning tasks, including Sudoku-Extreme and Maze-Unique, and demonstrate that increasing test-time compute along two axes (depth via more iterations and width via multiple stochastic trajectories) significantly improves accuracy.

**Compliance With Llm Reviewing Policy:**

Affirmed.

**Final Justification:**

See the revision below.

**Key Questions For Authors:**

1. The paper motivates the work using modern reasoning systems and test-time compute scaling. How well does the proposed attractor perspective extend to large language models or transformer-based reasoning architectures?
2. How does the proposed EqR framework fundamentally differ from prior work on Deep Equilibrium Models or energy-based reasoning models beyond empirical implementation details?
3. To what extent are the performance improvements due to the proposed attractor-based reasoning mechanism versus simply increasing test-time compute?
4. How robust is the proposed approach across a broader set of reasoning tasks? Please provide more experimental results in various benchmarks.
5. Have the authors evaluated whether the proposed methods remain effective under strict inference computation constraints?

**Limitations:**

The proposed framework is currently evaluated only on relatively small iterative reasoning models and structured synthetic tasks. As a result, it remains unclear whether the proposed attractor-based reasoning mechanism generalizes to larger models or real-world reasoning problems. Additionally, the greatest improvements rely on substantial increases in test-time compute, which may limit the practical applicability of the approach in compute-constrained settings.

**Strengths And Weaknesses:**

<Strengths>
- The paper presents an attractor-based interpretation of iterative reasoning models, framing inference as a dynamical system that converges toward task-conditioned fixed points. This perspective provides an interesting lens for understanding why additional test-time compute can improve reasoning performance.
- The work systematically analyzes two axes of inference scaling—depth (more iterations) and width (multiple stochastic restarts). The proposed framework provides useful insights into when such scaling is effective, depending on the structure of the learned attractor landscape.
- The paper introduces trajectory-level diagnostics that relate convergence residuals with task accuracy, offering a potentially useful way to analyze reasoning behavior in iterative models.
- The proposed algorithm, based on random initialization and path stochasticity, is simple to implement and appears to improve performance and stability in the evaluated benchmarks.
<Weaknesses>
- The motivation of the paper is framed around modern reasoning systems and test-time compute scaling, which are highly relevant in the context of large language models. However, the experiments are limited to relatively small iterative architectures and structured reasoning tasks (Sudoku and Maze). It remains unclear whether the proposed attractor perspective generalizes to large-scale reasoning models such as LLMs.
- The attractor-based interpretation appears closely related to existing work on Deep Equilibrium Models, energy-based models, and dynamical system views of neural networks. While the paper builds on these ideas, it is not fully clear what conceptual or theoretical advances distinguish EqR from these prior formulations.
- The experiments rely on two structured tasks that differ substantially in reasoning structure, like constraint satisfaction and path planning. Because of this difference, it is difficult to isolate whether the observed improvements arise from the proposed attractor-based reasoning mechanism or from task-specific properties.
- The strongest reported performance improvements appear when scaling test-time compute substantially. This raises the question of whether the gains reflect improved reasoning capability or simply increased search through larger inference budgets.
- The two main methodological contributions—randomized initialization and path stochasticity—are relatively simple modifications that resemble regularization or exploration strategies in optimization. It is unclear whether these interventions fundamentally improve reasoning mechanisms or mainly stabilize training and inference.

---

> ### Author Rebuttal · Authors · 2026-03-30
>
> ### **[Q1] Extension to large language models**
>
> Thank you for the question. We agree that our current experimental scope is narrower than the broad framing in parts of the introduction may suggest. We will tighten the framing accordingly. This paper is about non-autoregressive iterative reasoners in the continuous latent space, rather than LLMs defined in the discrete token space, used as a controlled setting to study attractor dynamics and test time scaling.
>
> ### **[Q2] Comparison with DEQ and EBM**
>
> Thank you for the question. EqR is conceptually closer to DEQs than to EBMs: we parameterize an update operator and reason via the attractor dynamics it induces (DEQs), rather than parameterizing an explicit energy and optimizing for its minimizer (EBMs). Unlike classical DEQs, however, we do not repeatedly solve for precise equilibria during training. Instead, we use truncated, imprecise attractors, borrowing the landscape intuition of EBMs while avoiding explicit energy optimization or sampling.
>
> ### **[Q3] Performance improvements attributions**
>
> The gains are not just from more test-time compute. We see three drivers:
>
> - **Same budget:** EqR outperforms the baseline at equal inference cost.
> - **Scaling:** EqR benefits more as compute increases beyond the training regime.
> - **Efficiency and selection:** EqR reaches a target accuracy with fewer steps and makes convergence-based selection reliable.
>
> As shown in the **table [[link](https://www.dropbox.com/scl/fi/iv0678vlabgc1a9oceoh8/table-scaling-compute.pdf?rlkey=359houxryu5gipl8k0sx6aeav&st=9tqf3bb4&dl=0)]**:
>
> - **Same inference budget:** EqR is better than the baseline (Sudoku 84.1 → 86.4, Maze 45.7 → 82.2).
> - **More compute:** EqR benefits more from scaling. With 4× depth, EqR improves to 93.0 (Sudoku) and 88.9 (Maze), while the baseline reaches 90.0 and 57.0.
> - **Larger scale:** With 4× depth + 128× width, EqR reaches 99.8 (Sudoku) and 93.0 (Maze), versus 98.4 and 64.1 for the baseline.
>
> This difference is reflected in efficiency at matched accuracy. As shown in the **figure [[link](https://www.dropbox.com/scl/fi/rhip328a7wxt27v49et26/scaling-accuracy.pdf?rlkey=iyzm3g693iuyr3ct4r4d5iiny&st=gdmmm8g1&dl=0)]** and table below, to reach the same accuracy of 92.99, the baseline requires 240.9 NFEs, whereas EqR requires only 64 NFEs. With adaptive halting, this is further reduced to 21.2 NFEs.
>
> | Method | NFE needed |
> | --- | --- |
> | Baseline | 240.9 |
> | EqR | 64 |
> | EqR + ACT | 21.2 |
>
> So performance does not improve just because we use more compute. The training intervention changes the iterative dynamics so extra iterations help more. The intervention also makes convergence align better with correctness (**Fig.5 [[Link](https://www.dropbox.com/scl/fi/4fl46o05pajn8mxnslzwc/Fig5.pdf?rlkey=zj7jb9m5aniidz8hyzgxbts4x&st=5vkw69zu&dl=0)]**)
>
> - **Both aggregation methods improve under EqR** compared with the baseline.
> - **Convergence based aggregation becomes both more accurate and more efficient than majority voting** under EqR, which is not the case for the baseline.
>
> ### **[Q4] Broader Task**
>
> We agree that broader task coverage is important, and we do not intend to claim broad empirical generality beyond the settings studied in this paper. As an additional result during rebuttal, we evaluated EqR on Mini ARC [1]:
>
> | Model | Accuracy |
> | --- | --- |
> | HRM | 44.85 |
> | TRM | 48.35 |
> | EqR | 55.28 |
>
> EqR outperforms both HRM and TRM on this benchmark, suggesting that the observed benefit is not limited to Sudoku and Maze. At the same time, we agree that one additional benchmark is not sufficient to establish broad generality, and we will revise the paper to state this scope more carefully.
>
> [1] Kim, Subin, et al. "Playgrounds for abstraction and reasoning."
>
> ### **[Q5] When Compute is Limited**
>
> Under strict inference budgets, we can utilize a trained halting head to enable Adaptive Computation Time (ACT). ACT allows the model to allocate more iterations to harder samples and stop early on easier ones, substantially reducing average compute while maintaining comparable accuracy.
>
> As shown in the **table,** ACT substantially reduces inference cost across all budgets while maintaining similar accuracy. For example, at the standard budget $(D, B) = (16, 1)$, ACT reduces average NFE from 16.0 to 5.4, with accuracy changing only from 84.3 to 84.0. Similar reductions are also observed at larger budgets. So while higher peak performance can benefit from more test-time computation, the framework also supports adaptive computation for efficiency.
>
> | Budget (D, B) | ACT | Acc. (%) ↑ | Avg. NFE ↓ |
> | --- | --- | --- | --- |
> | (16, 1) | ✗ | 84.3 | 16.0 |
> | (16, 1) | ✓ | 84.0 | 5.4 |
> | (1024, 1) | ✗ | 96.1 | 1024.0 |
> | (1024, 1) | ✓ | 95.3 | 58.7 |
> | (64, 128) | ✗ | 97.9 | 8192.0 |
> | (64, 128) | ✓ | 97.4 | 1400.6 |

---

> > ### Author Rebuttal · Reviewer_J868 · 2026-04-04
> >
> > Thanks to the authors for the response and additional experiments on the new testbed. The performance analysis of EqR and the evidence regarding its computational efficiency are well-presented.
> > However, my primary concern remains unsolved. While the additional evaluation on Mini ARC shows improvement over existing algorithms, the overall results across Sudoku, Maze, and Mini ARC indicate that EqR’s performance is disproportionately high on Sudoku. The authors need to provide a more rigorous explanation of how the architectural design of EqR relates to this performance variance across different testbeds. Without a clear justification for why it excels so significantly in one specific task compared to others, the method appears to be a specialized solution optimized primarily for Sudoku rather than a general-purpose one. Therefore, I will maintain my current rating of weak reject.

---

> > > ### Author Response · Authors · 2026-04-08
> > >
> > > We thank the reviewer for the constructive feedback and follow-up question. We agree that the stronger absolute performance on Sudoku deserves a clearer explanation. We also agree that architectural inductive bias can affect absolute performance differently across tasks. However, this inductive bias is **not introduced by our methods**. For fair comparison, EqR keeps the backbone architecture unchanged and strictly follows the corresponding baseline design in each task setting. Our contribution is on the **learning and inference algorithm side**, while maintaining the architecture aligned with prior arts. We elaborate as follows:
> > >
> > > ### 1. **EqR does not introduce a new Sudoku-specific architecture.**
> > >
> > > We agree that different tasks may favor different architectural choices, as already observed in the original TRM paper [1]. That said, these architecture choices remained the same as the corresponding baselines rather than being introduced by EqR.
> > >
> > > For fair comparison, we strictly follow the baseline (HRM, TRM) in each task setting: Sudoku uses an **MLP token mixer (MLP-t)**, while Maze and Mini ARC use the transformer architecture with self-attention. This inherited design choice can affect the absolute performance of each benchmark and helps explain why Sudoku may reach higher absolute accuracy than other testbeds.
> > >
> > > **However, the model design is separate from our contribution.** EqR does not change this task-dependent inductive bias. TRM and EqR both use the same recursive backbone, with the same shared update block applied at every step. In compact form, both methods follow the same recursion
> > >
> > > $$\mathbf{z}\_{k+1} = F\_\theta(\mathbf{z}\_k, \mathbf{x}),$$
> > >
> > > where $\mathbf{x}$ is the input tokens, and $\mathbf{z}\_k$ is the latent reasoning state (as tokens), $F_\theta$ is the MLP token mixer or transformer blocks.
> > >
> > > EqR introduces training interventions rather than model architecture **(Pseudo Code [[Link](https://www.dropbox.com/scl/fi/5p9ypq90ko8qq3rb5mxcv/eqr-pesudo-code.pdf?rlkey=7ex8xgm8aky9swkq67r4aqxru&st=q5btsssm&dl=0)])**. Therefore, while task-dependent inductive bias may explain some cross-task variance in absolute accuracy, it is not evidence that our method is architecturally specialized to Sudoku.
> > >
> > > ### 2. **The improvement relative to baseline is not confined to Sudoku.**
> > >
> > > EqR consistently improves over the baseline on Sudoku, Maze, and MiniARC tasks.
> > >
> > > On Sudoku, EqR improves accuracy from the baseline TRM result of **84.1** to **99.8**. On Maze, the gain is even larger, improving from **45.7** to **93.0**. A similar trend also appears on Mini-ARC, where EqR reaches **55.28**, compared with **44.85** for HRM and **48.35** for TRM. Though absolute performance varies, these results show that our method is not a Sudoku-specific solution, but rather improves the quality of iterative reasoning more broadly across tasks compared to baselines.
> > >
> > > ### 3. **Our method also improves the transformer’s performance on Sudoku.**
> > >
> > > To further validate whether our method is tied to a particular (architecture, task) choice, we also test model design using self-attention transformers on Sudoku, therefore aligning the model architecture design with the other two tasks. The same training intervention and inference scaling improve model performance:
> > >
> > > | Method | MLP-t Acc | Transformer Acc |
> > > | --- | --- | --- |
> > > | Baseline | 84.1% | 72.0% |
> > > | + training intervention | 86.4% (+2.3%) | 74.7% (+2.7%) |
> > > | + inference scaling | 99.8% (+13.4%) | 95.9% (+21.2%) |
> > >
> > > These results support that the benefit of our methods is not tied to the MLP-t arch, but also transferred to self-attention transformers.
> > >
> > > ---
> > >
> > > In summary, we agree that inductive bias matters. The stronger absolute performance on Sudoku is better explained by the interaction between task structure and the inherited backbone choice following prior arts. EqR keeps the backbone fixed relative to the baseline and instead improves the learned recursive dynamics. The cross-task variance is therefore not evidence that EqR is specialized to Sudoku. We hope this addresses the reviewer’s comment on model architecture choices.
> > >
> > > [1] Jolicoeur-Martineau, Alexia. "Less is more: Recursive reasoning with tiny networks."

---

### Decision · Program_Chairs · 2026-04-30

**Decision:**

Accept (regular)

**Comment:**

This paper investigates the internal mechanisms of iterative reasoning models, hypothesizing that generalizable reasoning emerges when models learn task-conditioned attractors. The authors propose Equilibrium Reasoners (EqR), which intuitively train the model to correct for its mistakes. The method demonstrates improved reasoning accuracy and allows the models to reliably scale with increased test-time compute on tasks like sudoku and maze.

Reviewers found the attractor-based perspective to be an intuitive and conceptually valuable framework for understanding test-time scaling in iterative reasoners. Initial reviews raised valid concerns regarding the narrow evaluation scope on structured tasks, the lack of multi-seed statistical significance, and the possibility that the performance gains were derived from generic regularization rather than fundamental changes to the attractor landscape. In the rebuttal, the authors provide additional experiments to verify the generality of the method. Another key concern of the reviewers was some unjustified claims, which the authors revised during rebuttal.